# Substrate thermal properties influence ventral brightness evolution in ectotherms

Jonathan Goldenberg [1✉], Liliana D'Alba [1], Karen Bisschop [2,3], Bram Vanthournout[1] & Matthew D. Shawkey [1]

The thermal environment can affect the evolution of morpho-behavioral adaptations of ectotherms. Heat is transferred from substrates to organisms by conduction and reflected radiation. Because brightness influences the degree of heat absorption, substrates could affect the evolution of integumentary optical properties. Here, we show that vipers (Squamata:Viperidae) inhabiting hot, highly radiative and superficially conductive substrates have evolved bright ventra for efficient heat transfer. We analyzed the brightness of 4161 publicly available images from 126 species, and we found that substrate type, alongside latitude and body mass, strongly influences ventral brightness. Substrate type also significantly affects dorsal brightness, but this is associated with different selective forces: activity-pattern and altitude. Ancestral estimation analysis suggests that the ancestral ventral condition was likely moderately bright and, following divergence events, some species convergently increased their brightness. Vipers diversified during the Miocene and the enhancement of ventral brightness may have facilitated the exploitation of arid grounds. We provide evidence that integument brightness can impact the behavioral ecology of ectotherms.

[1] Evolution and Optics of Nanostructures group, Department of Biology, Ghent University, 9000 Ghent, Belgium. [2] Terrestrial Ecology Unit, Department of Biology, Ghent University, 9000 Ghent, Belgium. [3] Theoretical Research in Evolutionary Life Sciences, Groningen Institute for Evolutionary Life Sciences, University of Groningen, 9700 CC Groningen, The Netherlands. ✉email: jonathan.goldenberg@ugent.be

The evolution of organismal coloration depends on multiple ecological and evolutionary factors[1], and no single function can fully explain color variation across the animal kingdom. Most previous studies on animal coloration have focused on colors for camouflage and mimicry e.g., refs. [2–4] or on sexual selection or social signaling e.g., refs. [4–6]. However, pigmented integument can also have significant thermal effects[7,8]. By selectively absorbing and reflecting solar and environmental radiation, the pigmented tissue can directly affect body temperature[9,10]. The sun's energy-rich radiation spans from the UV-visible (300–700 nm) to the near infrared (NIR: 700–2500 nm). Brightness is the relative amount of light reflected from a surface, and it can have a strong effect on temperature because, all things being equal, a bright material absorbs less solar radiation than a dark material[11].

Solar energy can be transmitted directly or indirectly to an organism (Fig. 1). The former occurs via direct exposure to the sun's radiation, and the latter via transfer from the surroundings including the substrate. Indirect transmission can vary with the type of substrate on which the organism lives; for example, each soil type has a different specific heat capacity (i.e., $c_p$: amount of energy required to raise 1 kg of substance by 1 °C; ref. [12]) and therefore, soils with different heat capacities will store different amounts of heat. Heat is transferred by conduction (direct contact, i.e., by molecular interactions), convection (within fluids) or radiation (electromagnetic waves, without direct contact) mechanisms[13,14]. Soils convey heat to other objects mainly through conduction and secondarily via radiation (infrared energy). The immediate transfer of heat from low $c_p$ (water-limited) soils such as sand or rock to other objects is higher than in high $c_p$ substrates (water-rich) such as humid forest grounds[12], because low $c_p$ soils release heat faster. Thus, variation in soils' thermal properties could be particularly important for species such as squamates that live in continuous and close contact with their substrate. Because integumentary brightness in part determines the amount of heat transferred, its evolution could be influenced by the substrate on which an animal resides.

Brightness of the integument is largely determined by melanins, a ubiquitous class of multifunctional macromolecules[15]. Melanins can absorb and transform solar radiation into heat[16–18], and have good electrical conductive properties[16,19–21], in turn affecting thermal conductance[22]. Energy conduction occurs via the transfer of ions and electrons, and in melanin it increases in water-rich environments[20,23–26]. In reptiles, melanin is found in organelles called melanosomes that are in turn housed in

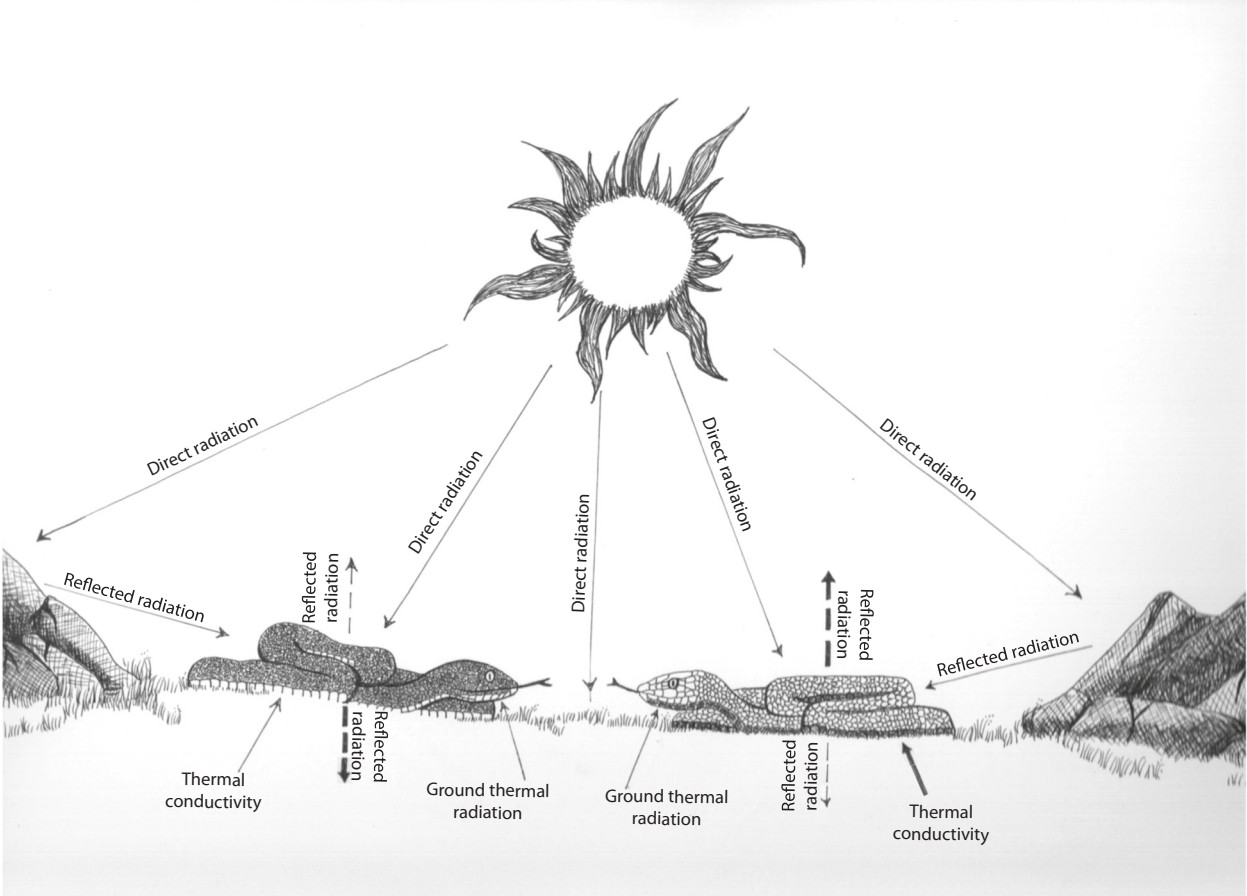

**Fig. 1 Flow of energy between direct-and-indirect solar radiations and the organism.** Here, we assumed convective heat exchanges (i.e., wind), evaporative cooling from metabolism, food uptake, and urine/faeces production to be constant. The formulated model is a simplification and presented in a conceptual manner to reflect our hypothesis. The direct solar radiation contains all energy-rich radiations, spanning from the UV-visible (300–700 nm) to the near infrared (NIR: 700–2500 nm). The reflected radiation and the ground thermal radiation contain IR energy. The ground thermal radiation and the thermal conductivity increase with the increase of direct solar radiation. However, different grounds present different specific heat capacities ($c_p$), which ultimately affect the amount of heat released by the substrate. Solid lines represent the energy received by the organism; dashed lines represent the energy reflected by the organism. Thicker dashed lines indicate high reflectance; thinner dashed line show low reflectance. Given the unique properties of melanin (see main text), we reasoned that a brighter integument will reflect more direct or indirect solar energy, while a darker integument will absorb more direct or indirect solar energy. Adapted from Porter & Gates[13] and Porter et al.[89]. Drawing: Karen Bisschop.

**Table 1 Data acquisition in ImageJ.**

| Body region | Region of interests (ROIs) | Description |
|---|---|---|
| Venter | 3 areas. Each spanning 1/10 of the trunk length[a] | Maximum outline within the ROI. Only ventral scales |
| Dorsum | 3 areas. Each spanning 1/10 of the trunk length[a] | Maximum outline within the ROI. Only dorsal scales. |
| Head | 3 areas | Maximum outline within the ROI. Only parietal (cranial) scale |
| Pattern | 3 areas | Maximum outline of pattern areas within the dorsum |

Our custom-made macro guides the user throughout analyses. Note that reference selection tool helps the user to stick to the reference length. For more details, please refer to Fig. S2
[a]We aimed for 1/10 for comparative purposes.

melanophores[27]. These unique thermal properties of melanins led us to predict that a less melanic (brighter) venter would be favored for animals that live on hot radiative and conductive substrates, because it would allow them to better dissipate transferred heat.

Previous research on squamates has primarily focused on dorsal coloration, perhaps because it is visually recognizable and exposed to direct solar radiation e.g., refs. [7,28–31]. Fewer studies have investigated the ecological significance of ventral coloration e.g., refs. [32–35], and only a handful have examined the link between it and thermoregulation[8,11,36]. Older studies investigated the effect of substrate use on thermoregulation[37] and mentioned the thermoecological significance of ventral color reflectivity[2,38,39]. However, none specifically examined the evolution of ventral brightness.

Here we use a comparative approach to investigate the macroevolutionary processes involved in shaping ventral brightness. We hypothesized that, while the dorsal, head and pattern brightness have evolved through tradeoffs between thermoregulation, protection, and camouflage, ventral brightness has been mainly driven by the $c_p$ of the substrates. We predicted that species inhabiting hot, and highly radiative and superficially conductive substrates (i.e., low $c_p$) would express less melanic (i.e., darker) ventral integument than those on high $c_p$ substrates. Moreover, as latitude in part determines how much of the sun's radiation is received by the organism, while also directly affecting transmission of energy to the substrate[40], we expected higher latitude species to express a greater melanic ventral integument than lower latitude counterparts. Furthermore, body size can have a strong effect on the overall thermal inertia of an organism[41], therefore we advanced that larger species will benefit from a brighter venter given the slower cooling rates relative to smaller species[11]. Finally, we predicted that high altitude species will display a darker ventral integument than low altitude organisms, as high altitudes are generally colder than lower altitudes, and a more melanic venter may confer a thermal advantage. Using vipers as study organisms, we combined ancestral state estimations and Bayesian mixed models to test these hypotheses.

## Results

**Study design and methodology validation.** We analyzed 126 taxonomically unambiguous viper species from 31 genera, covering ~35% of the family Viperidae (The Reptile Database[42]; accessed [July 2020]); Fig. S1). As direct spectrophotometry measurements of live vipers are logistically challenging, we retrieved brightness levels from 4161 images of these species from peer-reviewed articles, field guides, Google Images, documentaries and our own data. We removed images that were clearly over-or-under exposed. To account for different lighting and setup conditions, we took the following steps: (a) selected multiple (head (M = 8.37, SD = 2.09), dorsum (M = 8.64, SD = 2.08), venter (M = 4.50, SD = 2.69), and dorsal pattern (M = 7.53, SD = 2.78)) pictures/video frames per species, (b) assessed observer variability, (c) avoided over/under exposed areas, (d) assessed the relationship between brightness data obtained from

image analyses and spectrophotometry, (e) assessed the variability of image brightness within species, and f) verified the relationship between the visible (Vis), near infrared (NIR) and Ultraviolet (UV) spectra.

For (a) We examined all images in ImageJ 1.52i through a custom-made interactive plugin that enables the user to obtain brightness levels (see Methods for more information). Here we defined brightness as the mean of the RGB values[43].

For each species we calculated the mean brightness for each of the four body regions (head, dorsum, venter, and dorsal pattern (geometric shapes that contrast with the dorsal ground coloration)) (Table 1, Fig. S2). For (b), to assess the repeatability of the measurements, four observers independently analyzed 12 species (388 images), (c) with the directive to avoid shaded areas and flashed ("burned") regions. We found positive relationship between the observers (Obs1 vs Obs2: r = 0.94; R² = 0.88; p < 0.0001, Obs1 vs Obs3: r = 0.99; R² = 0.98; p < 0.0001, Obs1 vs Obs4: r = 0.99; R² = 0.98; p < 0.0001, Fig. S3), so only one proceeded with data collection. For (d), we verified the relationship between brightness values obtained through spectrophotometry from 29 living squamate species (including four viper species; Table S2) and brightness obtained through photographs. We found a significant positive relationship (r = 0.79; R² = 0.62; p < 0.0001, Fig. S4), a solid support for our method, particularly given that images were by necessity from different individuals to include intraspecific variation. For (e), we examined the variation of image brightness within same 29 squamate species. All standard deviations are moderate and similar across species (Fig. S5, Table S3), supporting the use of multiple images to retrieve an overall brightness mean per species. For (f), since 50% of the sun's energy-rich radiation is confined to the NIR region[44,45] and images account only for the Vis spectrum, we verified the relationship between these two spectral regions from spectrophotometry measurements on the same 29 living squamate species (Table S2). We found a strong positive relationship (r = 0.79; R² = 0.62; p < 0.0001, Fig. S6) as predicted, supporting the use of the vis spectrum as a proxy for the full spectrum. UV radiation also plays a role in the overall heating process of an organism, thus we verified the relationship between the UV range and the Vis spectrum on the same dataset. We found a significant positive relationship (r = 0.68; R² = 0.46; p < 0.0001, Fig. S7), further supporting use of the Vis range as proxy for the full spectrum.

**Bright ventra associate with low $c_p$ substrates.** We found that integumentary brightness and substrate type were associated in all models of ventral and dorsal body regions. However, environmental and morpho-behavioral variables differ depending on body region. Specifically, ventral brightness, in addition to being positively associated with body mass and negatively with latitude variables, is strongly negatively associated with substrate type (Tables 2, S4–7); species living on substrates with low $c_p$ (i.e., arid grounds) have significantly brighter ventral coloration 69% [61,78] than on any other substrate (Table S8, Fig. 2a).

**Table 2 Influence of variables on the integumentary brightness.**

| Body region | Substrate | Body mass | Distribution | Altitude | Day cycle | Polymorphism |
|---|---|---|---|---|---|---|
| Ventrum | −** | +** | −** | | | |
| Dorsum | −** | | | −* | −* | |
| Head | −** | | | −* | | |
| Pattern | ~** | | | | | |

Significant parameters influencing the brightness of the four body regions. Summary output from the ventral (Tab. S4–7), dorsal (Tab. S9–12), head (Tab. S14–17), and pattern (Tab. S19–21) MCMCglmm models. Two asterisks (**) indicate strongly supported parameters (variable present in all mostly supported models (DIC < 5) and has a cumulative Akaike weight of >0.75); One asterisk (*) less strongly supported (variable present in any of the mostly supported models (DIC < 5) and has a cumulative Akaike weight of > 0.75). "+" indicates a positive relationship between the variable of interest and the body region brightness. "−" shows a negative relationship between the variable of interest and the body region brightness. "~" displays no clear trend between the variable of interest and the body region brightness. Substrate: category from low-to-high substrate $c_p$ (specific heat capacity); Body Mass: continuous from small-to-large species; Distribution: category from low-to-high latitudes; Altitude: category from low-to-high altitudes; Activity Pattern: category from day-to-night activity patterns; Polymorphism: binary category ("yes","no").

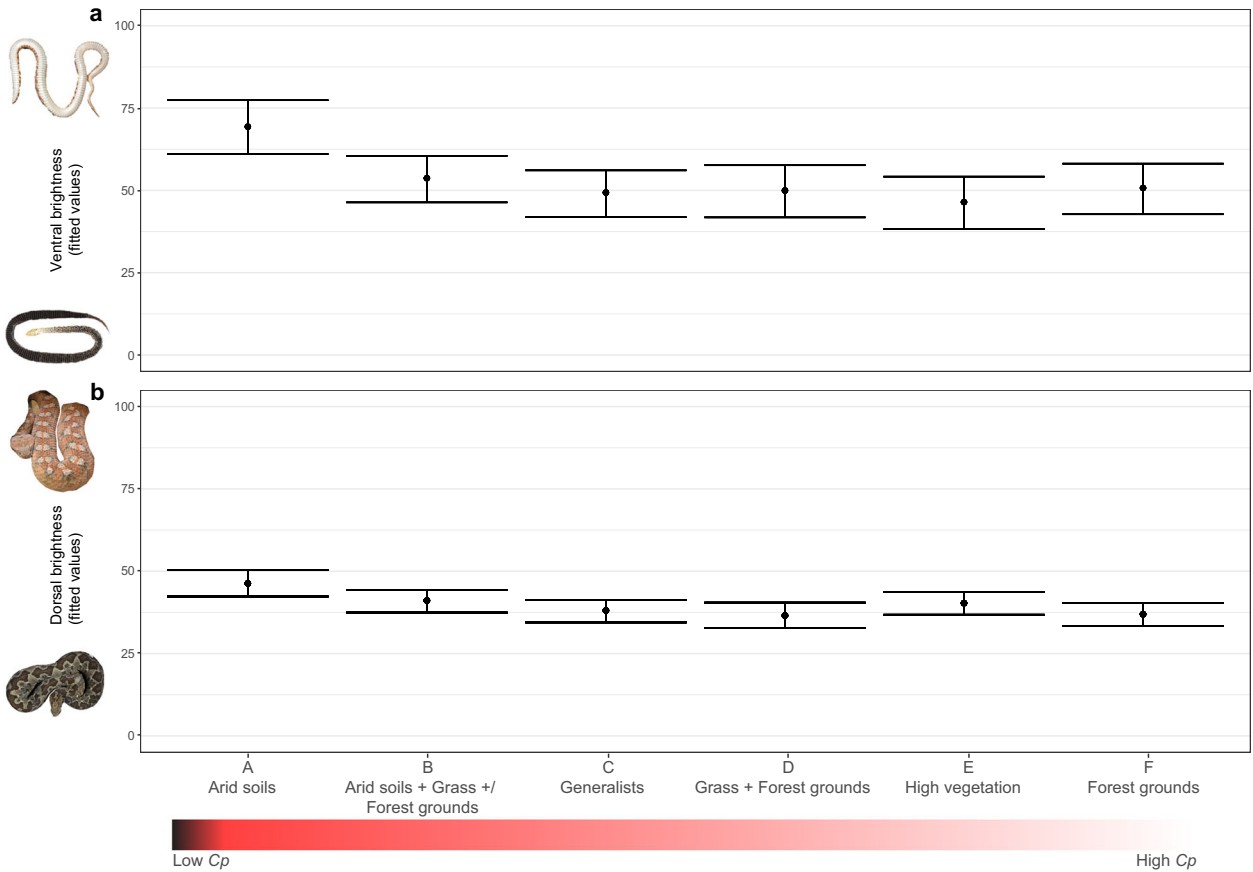

**Fig. 2 Brightness variation across different body regions and substrate types.** MCMCglmm-predicted values of ventral (**a**) and dorsal (**b**) brightness across the six different substrates. Bars represent 95% credible intervals. $c_p$ = Specific Heat Capacity. Generalists are species occurring on more than three different types of substrates. For a full description of substrate classification see Methods. Credits: *Echis leucogaster* (© Gabri Mtnez, moroccoherps.com—with permission), *Gloydius ussuriensis* (Orlov, N.L. et al.[90]), *Echis coloratus* (Wikipedia CC BY-SA 3.0), *Atropoides picadoi* (© Gert Jan Verspui, INaturalist, CC BY-NC 4.0).

Dorsal brightness is lower than the ventrum (Table S12, Fig. 2b) and it is negatively associated with substrate type, but, unlike the ventrum, also with activity pattern and altitude (Tables 2, S9–13, Fig. 2b). More specifically, diurnal and low altitude species are more likely to be brighter than nocturnal and high altitude animals (Tables 2, S12). Head brightness follows a nearly identical trend to dorsum (Fig. S8), but is only associated with substrate type and altitude (Tables 2, S14–18). The result is not surprising, as the brightness levels of two regions are strongly associated with each other (Fig. S10B). By contrast, pattern region is only associated with substrate type. (Tables 2, S19–21, Fig. S9).

**Evolution of brightness in different body regions.** The ancestral state reconstruction of the ventral brightness (Fig. 3a) suggests that the root viper node had a brightness level of 57% [44,71] 95% C.I., and that several groups convergently evolved bright or dark integument. During the mid-late Miocene (~14–6 Mya), brightness of *Echis* sp–*Cerastes* sp. (71% [59,84], *Pseudocerastes* sp.–*Eristicophis* sp. (64% [52,76], *Bitis* sp. (61% [50,72]), *Causus* sp. (58% [46,70]), *Daboia* sp. (58% [47,69]), and few species of *Crotalus* sp. (58% [51,64]), independently increased (Fig. 3a), and decreased in *Bothrops* sp. (53% [45,60]), *Gloydius* sp. (43% [33,52]), and *Vipera* sp. (45% [36,55]). The results are further supported by the Stayton's C1–C5 metrics of convergences based on dorsal and ventral

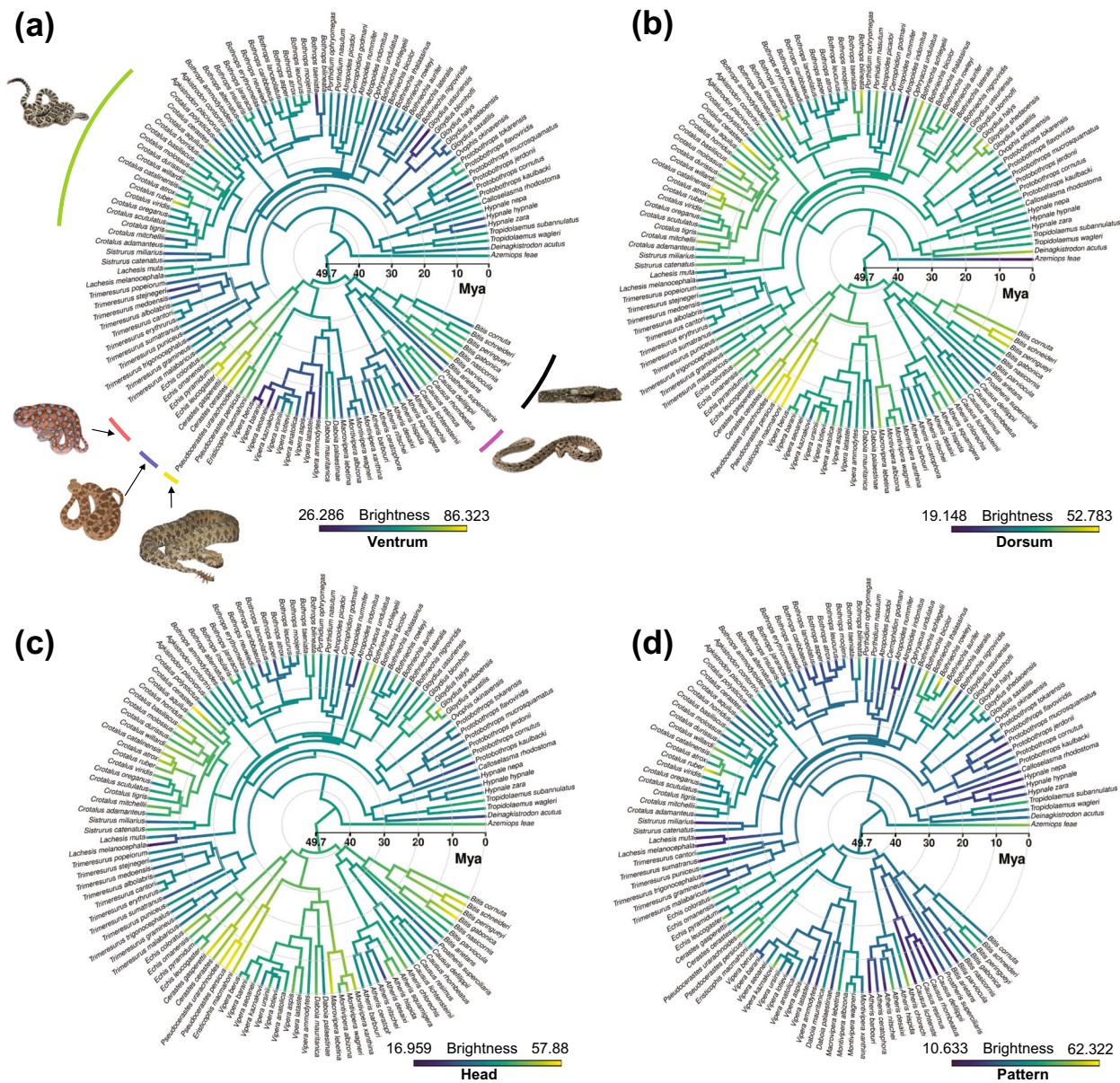

**Fig. 3 Viperidae ancestral brightness estimation across different body regions.** Viperidae ancestral state estimation of (**a**) ventral brightness, indicating convergent evolution of particularly dark and bright ventral integuments; (**b**) dorsal brightness showing an overall convergent brightness increase across different genera/species; (**c**) head brightness showing a similar trend to the dorsal brightness; and (**d**) pattern brightness indicating a global decrease of brightness since ~20 Mya. Note that six species (*Atheris squamigera*, *Trimeresurus albolabris*, *T. erythrurus*, *T. medoensis*, *T. popeiorum*, *T. stejnegeri*) do not show clear pattern shapes, therefore we dropped them from pattern analysis. Branches with blues (darker tonality) indicate dark colors, greens show the mid-brightness ancestral state and yellows (brighter tonality) specify bright colors. Credits: *Crotalus oreganus* (© Will Flaxington, CC BY-NC 3.0), *Echis coloratus* (© Matthieu Berroneau—with permission), *Cerastes cerastes* (© MinoZig, CC BY-NC 3.0.), *Pseudocerastes urarachnoides* (© Omid Mozaffari, CC-public domain), *Causus rhombeatus* (© Paul Venter, CC BY-NC 3.0.), *Bitis arietans* (© Jonathan Goldenberg).

brightness of the focal taxa (Table S23); specifically C1 (i.e., the maximum distance between two lineages that has been brought together by subsequent evolution) = 0.54 ($p = 0.00$) coupled with C5 (i.e., the number of convergent focal taxa that reside in a distinct region of the polymorphospace) = 15 ($p = 0.00$) show that the focal taxa significantly cluster together in a separate region of the polymorphospace driven by ventral brightness (Table S23, Fig. S14). As another line of evidence, the 95% C.I. phenogram, which projects the phylogeny in a space defined by the ventral brightness over time (Fig. S15B), shows that ventral brightness of the focal taxa shifted to bright brightness levels independently over time, corroborating our findings.

Brightness on the dorsum also increased across several genera (mostly different genera/species than those whose ventral brightness increased) from the late Oligocene to the mid-Miocene (~24–14 Mya) (Fig. 3b). The head brightness follows the same trend as the dorsum (Fig. 3c). Lastly, unlike all other body regions, brightness of patterns decreased starting in the early Miocene (~20–15 Mya) (Fig. 3d).

## Discussion
We hypothesized that evolution of ventral brightness in ectotherms confined to the ground is driven by the heat retention properties of their primary substrates. We used ancestral reconstructions and

multinomial models, with vipers as study organisms, to test this hypothesis. Our results show that, as predicted, bright ventral integuments are associated with arid substrates, i.e., low specific heat capacity substrates. This suggests that a bright ventral integument may provide an evolutionary advantage to species living on arid soils by more rapidly dissipating the heat transferred from the ground, thereby potentially avoiding overheating.

Ventral brightness was affected not only by the substrate, but also by latitudinal distribution and body mass. However, contrary to our predictions, altitude did not play a role in shaping ventral brightness.

Latitude in part determines how much the sun's radiation is received by the organism, while also directly affecting transmission of energy to the substrate[40]. Similarly to Moreno Azócar et al.[36], our results indicate that species closer to the equator are more likely to have brighter venters compared to species at higher latitudes. The authors suggested a potential thermoregulatory function because ventral melanism was significantly affected by cloudiness and minimum net radiation. We propose that darker ventra may provide a thermal advantage in lower energy-rich radiation zones, because melanin not only can absorb and transform solar radiation into heat[16–18], but also conduct energy[16,19–21], promoting thermal conductance[22].

Body mass affects thermal inertia in that larger species take longer to change body temperature[11,41]. Our findings show that larger species have brighter venters, possibly because of the slower cooling rates relative to smaller species[11]. These results suggest that substrate conditions, together with latitude and body mass, shape the ventral brightness evolution in our ectothermic group.

Finally, high altitudes are not only generally colder than low altitudes, but also have higher radiation. Therefore, unlike substrates at low altitudes, those at high altitudes likely experience contrasting abiotic factors (e.g., high solar radiation, low temperatures), that ultimately affect the substrate thermal properties. Our predictions did not account for the effect of solar radiation, but only temperature, at high altitude, but given our results we suggest that the brightness of the venter is not influenced by altitude.

Dorsal and head brightness were also negatively associated with the substrate type, but they were darker than the venters across all substrate types (Fig. 2). Moreover, altitude and, on the dorsum, the activity pattern, were also associated with the dorsal and head brightness, suggesting that brightness of those regions is driven by multiple competing pressures such as camouflage, thermoregulation and UV protection. Indeed, we found that diurnal and lower altitude species are more likely to have a brighter integument than nocturnal and higher elevation animals on any given substrate. Darker integuments may contribute not only to better camouflage, but also to thermoregulatory functions in colder environments (e.g., allowing the brain to reach optimal temperatures faster[46]) and UV protection (e.g., protecting the animal from higher radiations at higher altitudes[47]). The present results are further supported by Martínez-Freiría et al.[48] who found that darker dorsal colors in European vipers are associated with cold environments. Overall, our findings on the dorsal and ventral brightness suggest that these body regions are subjected to divergent selective forces.

Only substrate type was associated with pattern brightness, with no clear reduction in brightness with higher specific heat capacity substrates. Recently, Pizzagalli et al.[49] found that specific geometric shapes on the viper dorsa are associated with different ground habitats. Our results provide another line of evidence that patterns primarily evolved for a function other than thermoregulation, most likely camouflage.

Substrate properties in a given location can rapidly change following climatic shifts. In the last 50 My Earth experienced multiple climatic changes e.g., ref. [50]; for instance, during the Miocene (~23–5 Mya) it underwent regional aridification phases due to new orogenic formations and changes in air circulation[50]. Such environmental variations can significantly affect the performance and the bauplan of ectothermic organisms[51]. Vipers evolved from an ancestral form 49.7 Mya ca.[52] and evolved during these selective climatic fluctuations. The observed increase in ventral brightness in the mid-late Miocene (Fig. 3a), suggests that following aridification, brighter integuments may have enabled species inhabiting arid soils to more rapidly dissipate ground heat.

Integumentary brightness evolved differently in ventral and dorsal regions, further suggesting that it is uncoupled in these body regions. On the ventral side, five groups ((1) *Causus* sp., (2) *Echis* sp. - *Cerastes* sp., (3) *Pseudocerastes* sp. -*Eristicophis* sp., (4) *Bitis* sp., and (5) few members of *Crotalus* sp.) independently enhanced brightness of their ventral integuments during the mid-late Miocene (~14–6 Mya). Extant members of the African species inhabit the Sahara region. Zhang et al.[53] estimated that this desert formed 7–11 Mya. Interestingly, the rise of these four bright ventral genera (group 2, 3) coincides with this aridification. In parallel, the uplift of East Africa during the late Miocene may have enabled the rise of several *Bitis* spp. following the shrinkage of rainforests and expansion of open habitats[54,55]. During the mid-late Miocene, wooded-savannahs gradually replaced tropical forests in Southern Africa[56], creating new vacant niches that could be filled by species with brighter venters such as *Causus* sp. Similarly, ancestral forms of *Crotalus* sp. rapidly radiated when the great mountain ranges formed in North America, rapidly aridifying the surrounding environment[57]. On the other hand, the steep decrease in brightness experienced by the *Gloydius* complex, especially *G. ussuriensis*, may be linked to the uplift of the Tibetan plateau, which provoked an increase in precipitation in the loess plateau/east Asia around 8–9 Mya[58] producing forests in which dark ventral species may perform better.

Brightening of the dorsum and head may have been a response to the gradual replacement of forests with open areas since 20 Mya ca.[59], where brighter integuments may have provided an evolutionary benefit for camouflage and thermoregulatory purposes.

In contrast to any other body region, the pattern area follows an overall decrease of brightness starting from 20 Mya. Darker and contrasting patchy areas across the dorsum may have provided an evolutionary advantage by disrupting the ground coloration to better blend with the surrounding environment[49].

While the dorsum is exposed to both solar radiation and prey/predator sight, the ventral region is mostly cryptic in species confined to the ground. Ectotherms are highly susceptible to the surrounding environmental conditions to attain body temperatures that maximize performance. Thus, different brightness levels on different body regions may confer specific advantages to achieve the desired function[8,34]. Our results support the hypothesis that brightness of exposed body regions is not only selected for thermoregulatory properties, but also for protection and camouflage[29,60]. In contrast to Smith et al.'s[8] prediction that ventral color reflectivity would have little effect on thermoregulation, as the animals lie flat against the surface, we found that ventral brightness evolution appears to be mainly shaped by the substrate type. Hence, the upper and lower body regions experienced contrasting selection pressures, leading to the current variation in extant viper color brightness.

We have here provided evidence for the significance of brightness of an often-neglected body region, and laid the groundwork for future studies examining the reflectance, emissivity and thermoregulatory properties of melanin. Moreover, it may be important to implement such results in climate change

risk assessments due to the potential impact they can have on species distribution.

## Methods

**Species selection and dataset construction.** Vipers (Viperidae Oppel, 1811) are a family of venomous snakes that evolved 50 Mya c.a.[52,61]. Unlike other snakes, vipers use a sit-and-wait foraging behavior e.g., ref. [62], and therefore their substrate type likely plays an important role in regulating their body temperature. To date, 365 viper species (The Reptile Database[42]; accessed [July 2020]) are distributed across the globe ranging from the tropics to the higher latitudes[63] (>60° N). The observed large diversity, coupled with their feeding strategy and a relatively long evolutionary history, makes this family an ideal study organism to investigate how ventral brightness evolved under divergent selective environments.

To determine whether the specific heat capacity ($c_p$) of substrates has played a role in the evolution of ventral brightness, we extracted information on integument brightness levels of 126 taxonomically unambiguous viper species from 31 genera distributed across the world. We previously explained in Results section our precautions to retrieve brightness levels in different species from images and we refer the reader to that paragraph. All the references to the collected 4161 images are available at the provided repository.

As integumentary brightness may be evolutionary constrained across different regions of an organism, we took measurements from three regions of interest in each of the four body parts, i.e., head (M = 8.37, SD = 2.09), dorsum (M = 8.64, SD = 2.08), venter (M = 4.50, SD = 2.69), and dorsal pattern (M = 7.53, SD = 2.78; geometric shapes that contrast with the dorsal ground coloration) (Table 1, Fig. S2) and for each species we calculated the mean brightness for each body section. Clear images of the ventral side are rare, hence we analyzed a lower proportion of images for this cryptic body region. Six species (*Atheris squamigera*, *Trimeresurus albolabris*, *T. erythrurus*, *T. medoensis*, *T. popeiorum*, *T. stejnegeri*) do not show clear pattern outlines, therefore we dropped them from any pattern analysis.

Soils display different $c_p$ based on the amount of water they contain[12]. Arid substrates (low $c_p$) quickly dissipate the heat via conduction and radiation, while forests grounds (high $c_p$), slowly absorb and diffuse the heat. Accounting for direct $c_p$ values measurements on different soil types (Table S1), we classified substrates based on a $c_p$ gradient, i.e., from low (arid substrates such as deserts and rocky soils) to high (forest grounds). We looked in the literature to find the substrate and the ecological context for each species (all references are available at the provided repository). We identified six *Substrate* categories based on the aridity and ground composition; from low to high $c_p$: A = arid substrates (e.g., sandy, rocky grounds); B = arid substrates in combination with forest grounds and/or grass patches; C = species inhabiting more than three different substrates; D = grass patches in combination with forest grounds; E = arboreal (high vegetation) species; and F = forest grounds. A (n = 13), B (n = 30), C(n = 25), D (n = 15), E (n = 21), F (n = 22). Some categories may display similar or overlapping substrate $c_p$ values. For example, category B and D both present grass patches and/or forest grounds. However, species falling within category B will likely have to endure higher ground heat stresses than those on category D, because they also exploit arid substrates. For a detailed description of our substrate classification, please see Table S1.

We also incorporated other environmental data that could significantly affect the amount of the sun's energy-rich radiation received by the animal and substrate[64,65], and morpho-behavioral information that can influence thermal heat transfer (for a frequency table please see Table S22): (1) Altitude range: as species are distributed across a vast altitude range and there is very little information available on density distribution across the species ranges, we followed the thresholds provided in the mountain system classification of Körner et al.[66], and produced the following five categories: Low (x ≤ 500 m), Low-Medium (x ≤ 1000 m), Medium-High (500 m < x ≤ 4000 m), High (1000 < x ≤ 4000 m), All (all the range). In this study, the threshold of 4000 m corresponds to the highest examined viper distribution (i.e., *Gloydius halys*[67]). Furthermore, vegetation-based zonation at lowlands are very susceptible to climate conditions[68], thus different regions experience different upper limits for vegetation zonation, in turn potentially affecting the heat exchange dynamic between substrate-organism. Therefore, we increased the lowland elevation from 300 m (proposed by Körner et al.[66]) to 500 m as the latter reflects an average of the lowland thresholds (upper elevation limits for lowland zonation are 600–700 m[69]). Finally, some viper species can span across a vast altitudinal range (e.g., *Atropoides picadoi*[70]), while others are restricted to a well-defined zone (e.g., *Bothriechis rowleyi*[71]). Therefore, a broadly distributed species may overlap the distribution of a limitedly distributed one. However, the species that spans across a vast range will likely be exposed to more selection pressures due to different levels of abiotic factors, such as humidity, temperature and solar radiation, that can ultimately affect the species thermal balance. Consequently, in this study we produced the above-mentioned altitude categories that also account for the species' ecology. A graphical representation of altitude distribution across species is in Fig. S10; (2) Distribution: the studied species are distributed across all latitudes, but, similarly to elevation range, very little information is available on density distribution across the species ranges. Therefore, we classified the latitudinal distribution following the radiation index presented in Barry & Chorley[40]: Tropical, Subtropical, Temperate, Temperate-

Polar or any combination thereof. The level Temperate-Polar presented only one species (*Vipera berus*); however, we kept that in our analyses as it did not affect model convergence. Finally, latitude is linked with solar radiation, in turn affecting temperature. Our models, by including the latitudinal distribution of a species, account for the effect of different solar radiation levels, and thus, indirectly, temperatures, on shaping the physical characteristics of the local substrate; (3) Body mass: from Feldman et al.[72]; (4) Polymorphism: species displaying intraspecific variation and/or sexual dimorphism were classified as polymorphic; and (5) Activity pattern: behavioral trait defined as Diurnal, Nocturnal, Both or Unknown (all references used to score and classify the analyzed variables are available at the provided repository). Vipers can ontogenetically shift at morphological (e.g., color brightness shift in *Tropidolaemus wagleri*[73]) and behavioral levels (e.g., juveniles of *Bothriechis lateralis* live on the forest floor whereas adults are arboreal[74]). Hence, we primarily analyzed adults for our classification. However, if adults and juveniles occupy the same niche and do not display any color brightness shift, we included both stages. We did not account for tail brightness as it can be used for luring purposes and/or as a prehensile tool[75,76].

**Brightness quantification.** We analyzed all images in ImageJ 1.52i through a custom-made interactive plugin (MacroBright v.0.1)[77] that guides the user to select the region of interests from which the researcher can obtain the brightness levels (.json file available at the provided repository).

First, the user loads a folder with images and provides, image-by-image, a reference scale, and then outlines the maximum area of the region of interest. The obtained RGB values are then saved in.csv format (Table 1; Fig. S2).

**Spectrophotometry.** To quantify the reflectance of the 29 squamate species (Table S2; reptile collection of Tel Aviv University's Garden for Zoological Research), we deployed a dual spectrophotometer and light source (AvaLight-DH-S Deuterium-Halogen Light Source and AvaLight-HAL-(S)-MINITungsten Light Source) setup (Avantes Inc.,Broomfield, CO, USA across the UV-Vis-NIR range (300–1030 nm) connected to a bifurcated fibre optic cable and held at 90° using a RPH-1 probe holder. To account for measurement repeatability, we acquired three spectra from three selected points on the trunk (Fig. S13). Because no invasive procedures were performed on live animals, data collection was performed under the supervision of the reptile keeper.

**Statistics and reproducibility.** We used the species-level viper phylogeny by Alencar et al.[52], which is based on 11 genes (six mitochondrial and five nuclear) and 1186 sequences from 263 taxa, and to date is the most complete reconstruction of this family. We conducted all analyses in R v.3.6.2[78]. As a first step, we matched the phylogenetic tree to our dataset (i.e., 126 species) with the treedata function in "geiger"[79]. For our multinomial analyses, we employed a Bayesian approach that allows us to interpret our results in terms of posterior probabilities.

**Body brightness evolution.** To investigate the evolutionary history of integumental brightness we estimated ancestral states of color brightness from each different body region using "phytools"[80]. We used a model-based approach to map the estimating states at internal node using maximum likelihood with the contMap function. Then, to assess the observed convergences for ventral brightness among focal taxa (*Bitis parviocula*, *Bitis peringueyi*, *Causus resimus*, *Causus defilippii*, *Daboia mauritanica*, *Eristicophis macmahoni*, *Pseudocerastes urarachnoides*, *Pseudocerastes persicus*, *Cerastes cerastes*, *Cerastes gasperettii*, *Echis pyramidum*, *Echis omanensis*, *Echis coloratus*, *Echis leucogaster*, *Crotalus ruber*, *Crotalus cerastes*) we employed "convevol"[81] to estimate the convergent metrics, where C1–C4 are distances and C5 is a frequency-based degree of convergence[82]. As head brightness is highly correlated with dorsal brightness (Fig. S10B), pattern brightness displays a reduced dataset (*Atheris squamigera*, *Trimeresurus albolabris*, *T. erythrurus*, *T. medoensis*, *T. popeiorum*, *T. stejnegeri* do not show clear pattern shapes, therefore we have to remove those species for any analysis involving pattern brightness), and our interest relies on understanding the convergence patterns of integument brightness, we defined a polymorphospace using ventral and dorsal brightness. Finally, to further support the convergence pattern in ventral brightness, we produced a phenogram with 95% confidence intervals using "phytools"[80].

**Brightness of body regions and substrate association.** To assess the probability that a viper will display a specific brightness in a given substrate, we performed multiple Markov chain Monte Carlo Generalised Linear Mixed Models (MCMCglmm[83]) while accounting for phylogeny using the pedigree command. Initially we wanted to verify if substrate type was a significant explanatory variable to describe brightness on different body regions. We first constructed four different global models, one for each body region, setting every time the brightness of the body region of interest as the dependent variable and verified which variables better explained variation in the system using the dredge function in "MuMIn"[84] ranking by Deviance Information Criterion (DIC). We defined variables as strongly supported if they were present in all mostly supported models (DIC < 5) and had a cumulative Akaike weight of >0.75[85,86]; less strongly supported if they were present

in any of the mostly supported models (DIC < 5) and had a cumulative Akaike of >0.75. Then, to retrieve the posterior means of the significant parameters, we set those as fixed effect for each body region model. All mostly supported models reported substrate type as a significant predictor. Therefore, as our research question is to predict brightness levels on different body regions in response to different substrates, to produce our graphical outputs we set substrate type as fixed effect and to account for the other significant variables from the global model we set those as random. The construction of these graphical models was a necessary step in order to extract the probability intervals of only the substrate types. We ran all models setting 1000000 MCMC iterations, burnin = 40000 and thin = 20. The phylo object was converted to ultrametric values through the force.ultrametric function available in "phytools"[80]. As we did not have a prior knowledge of how brightness is related to different substrates and because all models converged (see below), we defined the default priors[87], and family (a character vector describing the traits class) was set as "gaussian". Then we extracted the probabilities of the fixed effects through the function predict. Finally, we plotted the results in "ggplot2". To assess convergence, we ran five models to verify if they converged to the same posterior distribution. The Gelman and Rubin criterion[88] shows that the point estimate values of the variables of interest are all confined between 1 and 1.02, and the multivariate psrf is 1. The results of the Gelman and Rubin criterion, coupled with trace plots, indicated that our model successfully converged.

**Reporting summary**. Further information on research design is available in the Nature Research Reporting Summary linked to this article.

## Data availability
All datasets are deposited at the provided repository[77].

## Code availability
All R-scripts and the macro developed for image analyses are available through our repository[77].

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

## Acknowledgements

We are grateful to the Tel Aviv University's Garden for Zoological Research for providing access to their reptile collection; especially to Shai Meiri, Ron Michlin, and Yossi Yovel for arranging the visit and Barak Levi for the great support during animal handling. We are thankful to Lionel Hertzog, Rafael Maia, and Joshua W. Lambert for the statistical support and to EON and TEREC groups for the multiple constructive discussions. We thank Florian Van Hecke and Bram De Vilder for the repeatability analysis. J.G. was funded by the Special Research Fund of Ghent University (BOF). K.B. was funded through a VICI grant (VICI grant no. 865.13.00) and the Special Research Fund (BOF) of Ghent University. This work was supported by the Research Foundation-Flanders (FWO) grant GOG2217N.

## Author contributions

J.G., L.D.A., K.B., B.V., and M.D.S. contributed to the writing of the manuscript and the interpretation of the results. J.G., L.D.A., and M.D.S. conceived the project. J.G., L.D.A., and K.B. performed the analyses.

## Competing interests

The authors declare no competing interests.
