## [Peer Review File · Communications Biology]

Reviewers' comments:

Reviewer #1 (Remarks to the Author):

This study aims to test whether ventral colouration in vipers is associated with thermal properties of the substrate, and it deduces an evolutionary history for colouration. Unfortunately, I found the study unconvincing because the reasoning often seemed confused, and the methodology does not seem robust. I made comments below to explain my hesitations in detail.

50-54 please clarify what 50% refers to. In terms of heat transfer it would make most sense to focus on energy, which is concentrated at short wave lengths. The colour of an object only refers to the visible light spectrum, and not to the UV wave lengths which contain most energy and would therefore be principally responsible for heat transfer. It seems therefore that the argument here is incomplete: colour may indicate absorption of relatively low-energy visible light, but what about the high-energy UV light which would be most important for thermoregulation? Experimentally, this could be resolved by measuring heat transfer for different light spectra (e.g. visible only, UV only, visible + UV etc) but I don't know if that has ever been done.

64 please clarify "effect of thermal energy is stronger": effect on what and stronger than what?

68-70 this conclusion does not follow logically from above: the rate of conduction is not dependent on the colour of the objects that conduct heat.

72 thermal energy = heat, and this sentence is a tautology

84+ the argument seems to be quite confused: absorption of solar radiation is limited to the surfaces that are exposed to it (i.e. dorsal and lateral surfaces), and it has nothing to do with ventral colouration. Melanin does not confer colouration other than light/dark, and the colouration of animals is principally conferred by carotenoids and surface properties. The chemical properties of melanin make it a good conductor, and a ventral surface high in melanin may have a greater conductivity than one low in melanin (note however that this would need to be verified experimentally), but this would be totally independent from any optical properties, because heat transfer between the ground and the animal does not depend on light. The conflation between different processes - melanin, light, colour, and conduction - make the rationale for the study unconvincing, at least as it is written now.

108-117 this method seems very unreliable given the great variation in image quality across these sources. Even the procedures that attempt to correct for this variation are very subjective, and there is likely to be pronounced variation between individuals within species. I could imagine a procedure where relative brightness is determined (e.g. difference between ventral surface and brightest area) may be less biased, but ultimately these data need to be measured directly following an established experimental protocol. It is very tempting to mine publicly available resources because of the logistical ease, but the end result is often of poor quality and does not answer the question.

132-136 short-wave UV radiation contains most energy, and IR the least: this section needs more explanation: what is the purpose of this comparison?

127-131 how many individuals per species were used? Did this comparison include the breath of sources of photographs used in the final analysis? There is still 60% of variation left to explain in this comparison.

190+ the discussion needs to be re-thought. For example, on lines 204-209 it is implied that melanin on the ventral surface of the animals absorbs solar radiation - this seems far-fetched. Also, the analysis presented here is correlational only, and the more mechanistic arguments need

to be moderated.

289-290 not really, they can also sit and wait in the sun and be exposed to other heat transfer processes than with the ground only

294 please clarify "selective environmental pressures": whether or not an environment exerts selective pressures need to be determined and cannot be assumed

296 is skin brightness related to melanin content: it is the chemical characteristics of melanin and not the optical properties that would influence conductivity

307-308 this sentence is incomplete

316-318 more information is needed here: how exactly was substrate type for each species determined? I cannot access the document on Google Drive.

Reviewer #2 (Remarks to the Author):

Review of "Substrate thermal properties influence ventral color evolution in ectotherms"

The evolution of color patterns across a diverse array of taxa has linked to multiple selective functions including, crypsis and camouflage, signaling, sexual selection and mimicry. In this study, an alternative explanation for the evolution of coloration is proposed. The authors hypothesize the thermal properties of the substrate and in particular the specific heat capacity (cp) of soils has been a major influence on the evolution of ventral coloration. One prediction emerging from the hypothesis is that species inhabiting hot, high radiative substrates with high conductivity should have lower amounts of melanin. Species in the snake family Viperidae were used to test their hypothesis and evaluate the prediction. Images from 126 viper species were obtained from various sources to derive an index of brightness. In addition, spectrophotometric data were obtained from 26 live species. Six explanatory variables were included in the analysis, substrate (based on soil type), body mass, distribution, elevation, activity pattern (i.e., nocturnal vs. diurnal activity), and presence of polymorphism. The analysis involved an ancestral character reconstruction using likelihood, and a phylogenetically informed analysis to determine the association between brightness and the six explanatory variables. Brightness data were determined for four body regions, ventrum, dorsum, head, and pattern. The results showed convergent evolution of ventral brightness. The MCMCglmm analyses found that brightness was associated with species living in arid conditions, with large body mass and distributions near the equator. A key conclusion is that ventral coloration of Vipers is influenced by the specific heat properties of their substrate preferences.

General Comments.

I think the title should be changed somewhat. The analysis does not evaluate the factors affecting the evolution of color patterns, but rather the amount of melanin. The analysis focuses on brightness rather than other elements of color, e.g., hue, chroma. In addition, the presentation of tables does not provide the reader with the ability to determine the amount of variation explained by each variable in the MCMCglmm analysis. Although substrate is significant, how much of the variation is associated with this predictor? The core table is only a matrix of + and - symbols and not how much variation each predictor variable contributed to the model. The conclusions do not include alternative potential explanations, such as background matching. For example, the Rock Rattlesnake has two morphs (Light and Dark). The morphs are associated with substrates that

differ in color (See Farallo and Forstner 2012 PLoS One).

Specific comments.

1. Line 21. Replace belly with ventral.
2. Line 22. Primary factor is substrate type, but an additional 2 variables are included. I would revise the sentence. Also, why leave out the potential explanation of background matching?
3. Lines 25 – 26. Revise to say that the analysis suggest the ancestral condition was...
4. Line 26. Potentially replace medium-bright with intermediate-bright.
5. Line 30. The analysis does not provide details about colors only levels of brightness. Revise.
6. Line 46. Replace colored tissues with pigmented integument.
7. Line 62. I would also cite Porter and Gates, since these authors described in detail the heat balance of an ectotherm.
8. Lines 71 – 95. The paragraph discusses melanin, which is involved in darkening rather than coloration. This argues for a change in the title.
9. Line 78. Should be Migliaccio.
10. Line 95. The paper focuses on brightness, but how would one classify a Viper that had a complete ventral color of red? The authors need to provide details about how chroma is embedded in their brightness index.
11. Line 100. Does background matching fall under the category of camouflage?
12. Line 108. Appropriate citation for Reptile Database is:
Uetz, P. Freed, P. and Hošek, J. 2019 The Reptile Database <http://www.reptile-database.org>.
access [add date]
13. Lines 109 – 111. Using photos/images from multiple sources is problematic. There is substantial processing of images if one is using jpegs. Please be clear about the types of image files used, the resolution, and file type (jpeg, raw).
14. Line 140. Is the index used color or simply brightness?
15. Line 143, Figure 2A. I would change the color scale to make brighter viperids as red (indicating hotter substrates) and darker viperids green (corresponding to cooler substrates).
In addition, the scale in Figure 2A shows that Crotalus was not much different in color than the ancestral condition.
16. Line 141 – 147. The 95% confidence intervals overlap all the mean values presented, which does not support an evolutionary trend. I regard this as evidence of status. Why not test for phylogenetic signal before engaging in these analyses?
17. Lines 162 – 167. I am confused about the description of the pattern. Based on the formula used in the analysis presented in the Supplementary Materials, the model was Ventral $\sim -1 +$ Substrate + Dorsal + Altitude + Distribution + Log body mass + Polymorphic + Day cycle. Substrate type does not positively influence body mass, it should read that Ventral coloration has a negative association with Substrate, but a positive association with body mass.
18. Line 169. I would substitute "elevation" for "altitude"
19. Lines 168 – 170. Is the interpretation that vipers inhabiting light soils have brighter backs? What about vipers that inhabit red soils and their dorsal color is red. Will the analysis verify this correlation?
20. Lines 260 – 261. Should be "to attain body temperatures that maximize performance."
21. Line 290. Uetz et al. havd 365 species.
22. Avoid starting a sentence with "This".
23. Line 293. Replace "path" with "history"
24. Line 294. Replace with "ventral colors evolved under divergent selective environments"
Supplementary Tables.
25. Line 300. The Google.docs link does not work.
26. Line 327. I would suggest replacing altitude with elevation.
27. Line 348. How were files with different image characteristics standardized.
28. Line 366. More details about the phylogenetic tree should be provide. How many taxa, was it ultrametric, etc.
29. Line 376. "contmap" is not a Bayesian ancestral reconstruction algorithm. I would use

"anc.Bayes:".

30. Line 399. Explain the selection of priors.

31. Grether et al. 2004 is not cited in the text.

32. Line 501. Italicize species names.

Line 567. There is a weird character in Ollala-Tarraga's name.

Tables S3, S8, S13, S18. What do the numbers in the first column represent? Add a column header that informs the reader the meaning of the numbers.

Tables S5, S6, S7, S10, S11, S12, S15, S16, S17, S20. Rather than using Substrate A, Substrate B, etc. Please provide the actual level, i.e., arid, forest, and so on. Otherwise indicate in the table legend what the substrate levels represent.

Reviewer #3 (Remarks to the Author):

I read the manuscript "Substrate thermal properties influence ventral color evolution in ectotherms" by Goldenberg et al. In general, I liked the manuscript, it flows when reading and the nature of the analyses are adequate. However, I found some aspects difficult to buy or I may have misunderstood part of the methods. I have to confess I am not a native English speaker; therefore, I will not correct language usage, but I marked some typos anyway.

Major Concerns.

The prediction made (line 100) looks vague or at least, it seems to pay attention to substrate only, there is no specific mention of elevation, which was analyzed and discussed as well as other abiotic and biotic aspects. Please, could you be more specific in the predictions in line with your results and discussion section?

Other major concern, perhaps the more difficult to digest is the scaling of "predictor" variables. For example, as far as I understood the types of Cp surfaces were coded. If so, the scale between different types of Cp substrates are equally separated and this may not be the actual case. I looked at the scale (Table S.1.) of different Cp substrate or materials and the definition of each one of the different "categories" you used. It might be possible that the values for sandy rocky grounds could be 0.22 and for arid substrates in combination with forest grounds and/or grass patches might be 0.30 (just guessing). Additionally, I can't see what would be the actual difference in Cp between arboreal (high vegetation) species and forest grounds, in any case the distance in terms of Cp between substrate A and B does not seem to be the same than that of E and F (Lines 319-323). I understand that getting the actual Cp values for each species is an impossible task, but I would like to see a cautionary comment on this respect at least.

The third major concern I have is with respect the authors used elevation. I see lots of overlaps and I find difficult to separate these values to understand what was done. I looked at the Figure S.10. and I see that some heights are forming kind of clusters, but I do not know how the authors used them in the analyses. My guessing is that the authors used the categories in lines 332-334 that lead me to think again about the amount of overlapping. Could the authors make this issue clearer?

Lastly, and understanding the complexity of the analyses carried out, I feel that despite the quantitative value of these variables (Cp and elevation) they are not independent of climatic variables, such as temperature. In my opinion even when sand Cp may be the same in light of the physical characteristics of sand; it is not the same a sandy cold desert (Patagonia, for example) than the Sahara sands in terms of heat dissipation for a snake. Could the authors include a paragraph considering this type of information?

I really liked the way the authors used color information from different sources and how they managed to input these data in a trustable and objective way. I also enjoyed the arguments about the electrical properties of melanin. I can see the authors really manage the subject.

Minor aspects.

Lines 118-127. For is written in boldface, should be regular font.

Line 136 Vis instead of vis.

I am not totally sure about the need of Figure 1. I might be wrong. Explain why the Figure must stay.

Revise references, there are some species names not in italics.

As mentioned before, I liked the way the manuscript is written and the focus the authors gave to it. I hope the authors could fix the major issues or at least make reasonable arguments to keep them and to explain the readers the limitations of using categories in the case of Cp substrates and elevation.

**Overview**

List of terms used in this document:

**Original article:** the first main article submitted

**Revised article:** the main article revised

**Original supplementary text:** the first supplementary file submitted

**Revised supplementary text:** the supplementary file revised

We thank all three reviewers for the insightful comments. We thoroughly analyzed all the concerns
and addressed them point-by-point. All changes in the revised texts (article + supplementary) are
highlighted in yellow, as well as our answers in this document to the reviewers. We first provide an
overview of major common concerns, and afterwards we address and answer point-by-point the
specific comments.

One of the main concerns raised in this revision relates to the use of the terms “brightness” and
“coloration” and how those terms are connected. In this study, we did not classify the integument
based on coloration, but rather based on brightness. We clarified this and use “brightness”
throughout the manuscript.

A second common concern regards the image brightness acquisition methodology, especially how
images from different sources were standardized. Images were retrieved from different sources
(lines: 133-134 revised article) and without knowing the camera specifications nor EXIF (e.g. AV, TV,
ISO information) data, it is not possible to directly standardize the pictures. However, our
methodology accounts for different lighting and setup conditions, and we present in lines 131-170 of
the revised article the precautions taken and our validation of the methods. A priori, we did not
collect images that were clearly over-or-under exposed. To provide a stronger support to our
methodology (in comparison to the original article), **1**) we included three more squamate species
(with respect to the original article) to the correlation analysis between spectrophotometry data and
image brightness ($r = 0.79$; $R^2 = 0.62$; $p = 3.5e-07$, Figure S.4 - line 156 of the revised article), and **2**)
we collected more images per species (mean = 9.81, SD = 1.41) which is then comparable with the
number of pictures per species used in the final analyses (previously we analyzed few images on the
ventral side (mean = 1.49, SD = 0.55), but now we focused on the dorsal, because ventral image
availability is very limited if not available at all for all those species –the same limitations we faced in
the final study. We report this in lines 360-366 of the revised manuscript); **3**) in addition to Vis-NIR
relationship ($r = 0.79$; $R^2 = 0.62$; $p < 2.2e-16$, Figure S.6 - line 165 of the revised article), we included a
further correlation analysis between Vis-UV ($r = 0.68$; $R^2 = 0.46$; $p < 2.2e-16$, Figure S.7 - line 169 of
the revised article); **4**) we asked two extra external observers to retrieve the image brightness from
the same subset provided to the second observer following the same directive of avoiding shaded
areas and flashed (“burned”) regions. With the addition of those two extra observers, the
correlation analyses strongly support the high repeatability of this image brightness collection (Obs1
vs Obs2: $r = 0.94$; $R^2 = 0.88$; $p < 2.2e-16$, Obs1 vs Obs3: $r = 0.99$; $R^2 = 0.98$; $p = 1.1e-10$, Obs1 vs Obs4:
$r = 0.99$; $R^2 = 0.98$; $p = 3.4e-09$, Figure S.3 – lines 150-152 of the revised article); **5**) we also verified
the variation of image brightness values within species, finding that all standard deviations are
moderate and similar across species (Figure S.5, Table S.3 – line 158-161 of the revised article).
The provided correlation results, coupled with the variability analysis on image brightness per
species and the repeatability analysis, support our image brightness collection methodology.

Finally, Reviewer #1 and Reviewer #2 could not access the provided Google link. We apologize for
the accessibility issue. We tested again the link via four different laptops running different operative
systems and browsers without any problem. Could you please try the link again?

["https://drive.google.com/drive/folders/15UMrWPMarHwDcz9GJ-i0sqWArdXAl0CZ?usp=sharing"](https://drive.google.com/drive/folders/15UMrWPMarHwDcz9GJ-i0sqWArdXAl0CZ?usp=sharing)
(perhaps a copy-paste of the last dot or extra space caused the problem).

**Main structural changes:**

- - Upon suggestion of Reviewer #2, we modified the article title from "Substrate thermal
properties influence ventral **color** evolution in ectotherms" to "Substrate thermal properties
influence ventral **brightness** evolution in ectotherms".
- - Figure 1 of the original article has been moved to supplementary upon suggestion of
Reviewer #3.
- - We added in the introduction of the revised article a diagram to explain our hypothesis as
clearly as possible.
- - Figure 4 of the original article has been revised. In the revised article we now provide a panel
with both ventral and dorsal brightness predicted values, so a reader can directly assess the
differences between the two. Specifically, dorsal brightness is notably darker than the
ventral brightness.
- - Several figures and tables have been added and/or revised in the revised supplementary text
to clarify concerns raised. All of them are indicated throughout this revision and the revised
article.

Below, we provide a point to point answer to all the comments raised.

**Reviewer #1 (Remarks to the Author): Evolutionary and ecological physiology, thermal biology,**
**heat transfer, herpetology**

This study aims to test whether ventral colouration in vipers is associated with thermal properties of
the substrate, and it deduces an evolutionary history for colouration. Unfortunately, I found the
study unconvincing because the reasoning often seemed confused, and the methodology does not
seem robust. I made comments below to explain my hesitations in detail.

**1)**

50-54 please clarify what 50% refers to. In terms of heat transfer it would make most sense to focus
on energy, which is concentrated at short wave lengths. The colour of an object only refers to the
visible light spectrum, and not to the UV wave lengths which contain most energy and would
therefore be principally responsible for heat transfer. It seems therefore that the argument here is
incomplete: colour may indicate absorption of relatively low-energy visible light, but what about the
high-energy UV light which would be most important for thermoregulation? Experimentally, this
could be resolved by measuring heat transfer for different light spectra (e.g. visible only, UV only,
visible + UV etc) but I don't know if that has ever been done.

50% refers to sun's energy-rich radiation. Nevertheless, we removed this sentence from the
introduction of the revised article and we report this value in line 161 of the revised article.

However, we must respectfully disagree with the reviewer regarding the comment that heat transfer
is mostly concentrated on short wavelengths (UV). Ultraviolet wavelengths have higher frequency
(and thus more energy) compared to longer (e.g. NIR) wavelengths. The result of this higher
frequency is that UV is strongly absorbed by the tissue, i.e. by organic molecules like proteins, lipids
and DNA. However, the net solar radiation absorbed by an animal spans all wavelengths from the
UV-Visible and NIR portion of the spectra (Stuart-Fox et al. 2017) and all these wavelengths have the
potential to be converted into heat and are indeed responsible for heat transfer. The UV range
accounts for only about 6% of the total solar radiation falling on the earth. Nonetheless, we now also

provide a correlation analysis between the Vis and UV spectra, showing a significant positive
relationship ($r = 0.68$; $R^2 = 0.46$; $p < 2.2e-16$, Figure S.7 - lines 168-170 of the revised article),
providing another line of evidence for the support of the Vis range as proxy for the full spectrum.

**2)**
64 please clarify "effect of thermal energy is stronger": effect on what and stronger than what?
We clarified this by rephrasing the speech in lines 61-63 of the revised article: "The immediate
transfer of heat from low c_p (water-limited) soils such as sand or rock to other objects is higher
than in high c_p substrates (water-rich) such as humid forest grounds (Farouki, 1981)".

**3)**
68-70 this conclusion does not follow logically from above: the rate of conduction is not dependent
on the colour of the objects that conduct heat.

As addressed in the overview section, we clarified the relationship between coloration and
brightness. Consequently, we do not refer anymore to coloration but only brightness. Specifically, in
the revised article we address this point in lines 65-67 as: "Because integumentary brightness in
part determines the amount of heat transferred, its evolution could be influenced by the substrate
on which an animal resides."

Right after this, the next paragraph introduces the pigment melanin and its optical and thermal
properties.

**4)**
72 thermal energy = heat, and this sentence is a tautology
We fully agree, and we clarified this by removing "thermal energy" in line 72 of the original article.
In the revised article it reads as: "This is particularly true because brightness of the integument is
largely determined by melanins, a ubiquitous class of multifunctional macromolecules (Prota, 1992).
Melanins can absorb and transform solar radiation into heat" - lines 68-70.

**5)**
84+ the argument seems to be quite confused: absorption of solar radiation is limited to the surfaces
that are exposed to it (i.e. dorsal and lateral surfaces), and it has nothing to do with ventral
colouration. Melanin does not confer colouration other than light/dark, and the colouration of
animals is principally conferred by carotenoids and surface properties. The chemical properties of
melanin make it a good conductor, and a ventral surface high in melanin may have a greater
conductivity than one low in melanin (note however that this would need to be verified
experimentally), but this would be totally independent from any optical properties, because heat
transfer between the ground and the animal does not depend on light. The conflation between
different processes - melanin, light, colour, and conduction - make the rationale for the study
unconvincing, at least as it is written now.

To clarify our research question and this concern, we produced a diagram showing our main
hypothesis (Figure 1 of the revised article). Solar radiation is not limited to the exposed surfaces.
Indeed, solar energy can be transmitted directly or indirectly to the individual. While the former is
achieved by direct exposure to the sun's energy-rich radiation, the latter occurs when substrates a)
reflect solar radiation, b) absorb solar radiation and they in turn radiate such energy. The degree of
atmospheric radiation will depend on the type of substrate on which the organism lives, where

different substrates may reflect more or less radiation. And finally as the reviewer mentioned, c)
conduction of heat (different substrates differently store the heat absorbed by the solar radiation).

As we do not refer anymore to coloration but only brightness, which is here used as a proxy for
melanin content, we have addressed this concern.

Finally, melanin can confer coloration (brown, grey, black through eumelanin, and yellow-red
through pheomelanin (even in reptiles e.g. Roulin, A. et al. 2013)).

**6)**

108-117 this method seems very unreliable given the great variation in image quality across these
sources. Even the procedures that attempt to correct for this variation are very subjective, and there
is likely to be pronounced variation between individuals within species. I could imagine a procedure
where relative brightness is determined (e.g. difference between ventral surface and brightest area)
may be less biased, but ultimately these data need to be measured directly following an established
experimental protocol. It is very tempting to mine publicly available resources because of the
logistical ease, but the end result is often of poor quality and does not answer the question.

The correction procedure may be subjective, but it is highly repeatable. In Figure S.2 of the original
supplementary file we show that two observers independently measured and scored a subset of 388
images from 12 species producing a coefficient of correlation of 0.98. Now, the Figure S.3 of the
revised supplementary file includes other 2 extra independent observers and the coefficients of
correlations are Obs1 vs Obs2: $r = 0.94$; $R^2 = 0.88$; $p < 2.2e-16$, Obs1 vs Obs3: $r = 0.99$; $R^2 = 0.98$; $p =$
$1.1e-10$, Obs1 vs Obs4: $r = 0.99$; $R^2 = 0.98$; $p = 3.4e-09$.

We have also added 3 species to the correlation analysis between reflectance measurement and
brightness of images (Figure S.4 of the revised supplementary text) and increased the image
sampling per species (mean = 9.81, SD = 1.41), providing a coefficient of correlation of 0.79, $p=3.5e-$
07 . Coupled with the new Figure S.5 and Table S.3 of the revised supplementary file that show a
contained variation in image brightness within species, our methodology provides strong support for
our analyses. Finally, our models are accounting for polymorphic species (i.e. species displaying
different brightness levels) – explained in lines 419-421 of the revised article.

**7)**

132-136 short-wave UV radiation contains most energy, and IR the least: this section needs more
explanation: what is the purpose of this comparison?

This follows our answer to comment (1). Although UV radiation has higher frequencies compared to
longer wavelengths, the UV range accounts for only 6% of the total solar radiation falling on the
earth and the IR ~55 % (NIR ~ 50%). Thus the contribution of UV to thermal heat will likely be less
significant.

**8)**

127-131 how many individuals per species were used? Did this comparison include the breath of
sources of photographs used in the final analysis? There is still 60% of variation left to explain in this
comparison.

We used 26 squamate species, and now we added three extra species, reported in Table S.2 of the
revised supplementary file. We changed in the revised article the number of species used too (line

154). With the addition of those species and increasing the image sampling per species (9.81 ± 1.41
SD), mimicking the numbers used for the final analyses, we obtain a coefficient of correlation of
0.79, thus explaining 62% of the variation. Finally, we retrieved the images for the comparison
analysis following the same methodology used in the final analysis.

**9)**

190+ the discussion needs to be re-thought. For example, on lines 204-209 it is implied that melanin
on the ventral surface of the animals absorbs solar radiation - this seems far-fetched. Also, the
analysis presented here is correlational only, and the more mechanistic arguments need to be
moderated.

By addressing the indirect effect of solar radiation in comment (5) we hope this point is clarified
now. Also, we believe we have been moderated in our arguments as we state in line 248-249 of the
revised and original article "We propose that darker ventra may provide a thermal advantage in
lower energy-rich radiation zones". We do not advance this as causation. Finally, as a closing
paragraph of the discussion section at lines 330-332 we propose future studies to investigate "the
reflectance, emissivity and thermoregulatory properties of melanin".

**10)**

289-290 not really, they can also sit and wait in the sun and be exposed to other heat transfer
processes than with the ground only

We agree and we removed the contentious "more" from the speech, and in the revised article, lines
346-348, it now reads as: "Unlike other snakes, vipers use a sit-and-wait foraging behavior (e.g.
Shine & Li-Xin, 2002), and therefore their substrate type likely plays an important role in regulating
their body temperature."

**11)**

294 please clarify "selective environmental pressures": whether or not an environment exerts
selective pressures need to be determined and cannot be assumed

As reviewer #2 suggested, we changed "selective environmental pressures" with "divergent selective
environments" – line 353 of the revised article.

**12)**

296 is skin brightness related to melanin content: it is the chemical characteristics of melanin and
not the optical properties that would influence conductivity

Melanin presence influences the brightness level. Therefore, given the relative amount of melanin in
the upper dermal layer, an animal will be brighter or darker and will play an important role on
radiation absorption and due to the chemical properties of melanin this will in turn influence the
overall heating potential of an individual.

**13)**

307-308 this sentence is incomplete

We modified the sentence in the revised manuscript at lines 365-366 as: "Clear images of the ventral
side are rare, hence we analyzed a lower proportion of images for this cryptic body region".

**14)**

316-318 more information is needed here: how exactly was substrate type for each species
determined? I cannot access the document on Google Drive.

We apologize for the accessibility issue. We tested again the link via four different laptops running
different operative systems and browsers without any problem. Could you please try the link again?
"<https://drive.google.com/drive/folders/15UMrWPMarHwDcz9GJ-i0sqWArdXAl0CZ?usp=sharing>"
(perhaps a copy-paste of the last dot may have caused the problem).

We also modified table S.1 in the revised supplementary text and we provide an in-depth
explanation of how we categorized the substrate type. Here after is what we write in the revised
article (lines 374-385): "We looked in the literature to find the substrate and the ecological context
for each species (all references are available at the provided repository). We identified six *Substrate*
categories based on the aridity and ground composition; from low to high c_p : A = arid substrates (e.g.
sandy, rocky grounds); B = arid substrates in combination with forest grounds and/or grass patches;
C = species inhabiting more than three different substrates; D = grass patches in combination with
forest grounds; E = arboreal (high vegetation) species; and F = forest grounds. A (n=13), B (n=30),
C(n=25), D (n=15), E (n=21), F (n=22). Some categories may display similar or overlapping substrate
c_p values. For example, category B and D both present grass patches and/or forest grounds.
However, species falling within category B will likely have to endure higher ground heat stresses
than those on category D, because they also thrive on arid substrates. For a detailed description of
our substrate classification, please see Table S.1."

In addition to that, in lines 14-32 of the revised supplementary text we write: "We categorized
substrates based on literature information (provided in the supplied repository) on habitat type and
ecology of species. Our classification reflects biologically relevant substrates on which the studied
species can be found.

Despite category E (high vegetation) and F (forest grounds) display similar substrate specific heat
capacity (c_p) values, the species ecology and the environmental pressures individuals experience on
those substrates are likely different. Therefore, for this analysis we separated those two categories.
Category B includes any of the arid substrates falling within category A plus any substrate falling
within category D. Category C includes at least four different substrate categories. We classified it
between category B and D because no matter which category combination is considered, the overall
heat stress from the ground is lower than the hotter substrates (i.e. A-B) and higher than the cooler
ones (D-F). A reader can argue that category C is redundant and should be incorporated within e.g.
category B or D. However, the selection pressures species have to endure in multiple substrates are
likely different from those who thrive on limited amounts, hence the addition of category C. As an
example, *Protobothrops mucrosquamatus* can be found on grasslands, forest grounds, shrublands,
around human settlements and agricultural lands, and have arboreal tendencies, thus we classified it
within category C (<https://www.iucnredlist.org/species/178409/7540882>;
<http://www.toxinology.com/fusebox.cfm?fuseaction=main.snakes.display&id=SN0111>).

We followed Dehgan, B. (2014) 5 m threshold height definition for a tree, whereas shrubs are
generally having a height of 1-5 m and multiple trunks."

**Reviewer #2 (Remarks to the Author): Evolutionary ecology, herpetology, evolutionary physiology**

Review of “Substrate thermal properties influence ventral color evolution in ectotherms”

The evolution of color patterns across a diverse array of taxa has linked to multiple selective
functions including, crypsis and camouflage, signaling, sexual selection and mimicry. In this study, an
alternative explanation for the evolution of coloration is proposed. The authors hypothesize the
thermal properties of the substrate and in particular the specific heat capacity (cp) of soils has been
a major influence on the evolution of ventral coloration. One prediction emerging from the
hypothesis is that species inhabiting hot, high radiative substrates with high conductivity should
have lower amounts of melanin. Species in the snake family Viperidae were used to test their
hypothesis and evaluate the prediction. Images from 126 viper species were obtained from various
sources to derive an index of brightness. In addition, spectrophotometric data were obtained from
26 live species. Six explanatory variables were included in the analysis, substrate (based on soil
type), body mass,
distribution, elevation, activity pattern (i.e., nocturnal vs. diurnal activity), and presence of
polymorphism. The analysis involved an ancestral character reconstruction using likelihood, and a
phylogenetically informed analysis to determine the association between brightness and the six
explanatory variables. Brightness data were determined for four body regions, ventrum, dorsum,
head, and pattern. The results showed convergent evolution of ventral brightness. The MCMCglmm
analyses found that brightness was associated with species living in arid conditions, with large body
mass and distributions near the equator. A key conclusion is that ventral coloration of Vipers is
influenced by the specific heat properties of their substrate preferences.

General Comments.

I think the title should be changed somewhat. The analysis does not evaluate the factors affecting
the evolution of color patterns, but rather the amount of melanin. The analysis focuses on
brightness rather than other elements of color, e.g., hue, chroma.

We agree and we changed the title from “Substrate thermal properties influence ventral **color**
evolution in ectotherms” to “Substrate thermal properties influence ventral **brightness** evolution in
ectotherms”. Also, as noted in the overview paragraph of this document, we clarified that we indeed
analyzed brightness and not color (i.e. hue) of species.

In addition, the presentation of tables does not provide the reader with the ability to determine the
amount of variation explained by each variable in the MCMCglmm analysis. Although substrate is
significant, how much of the variation is associated with this predictor? The core table is only a
matrix of + and – symbols and not how much variation each predictor variable contributed to the
model.

Unfortunately, the MCMCglmm output with more than one categorical fixed effect cannot return a
direct pMCMC for all levels as it automatically creates reference levels (even by suppressing the
intercept). So we explain in lines 466-473 of the revised article how we quantified the effect of each
fixed effect in determining the outcome variable (i.e. the brightness of one of the four body regions):
“We first constructed four different global models, one for each body region, setting every time the
brightness of the body region of interest as the dependent variable and verified which variables
better explained variation in the system using the dredge function in “MuMIn” (Barton, 2019)
ranking by Deviance Information Criterion (DIC). We defined variables as strongly supported if they
were present in all mostly supported models (DIC <5) and had a cumulative Akaike weight of > 0.75

(Marchetti et al., 2004; Buxton et al., 2017); less strongly supported if they were present in any of
the mostly supported models (DIC <5) and had a cumulative Akaike of > 0.75".
Specifically, in the caption of Table 1 of the revised and original article, we explain that red colors are
strongly supported variables and orange are less strongly supported variables. If we would make the
analogy with "significance" then red = highly significant, and orange = significant. No color indicates
no support, i.e. not significant.

The conclusions do not include alternative potential explanations, such as background matching. For
example, the Rock Rattlesnake has two morphs (Light and Dark). The morphs are associated with
substrates that differ in color (See Farallo and Forstner 2012 PLoS One).

Despite the Rock Rattlesnake (*Crotalus lepidus*) was not analyzed in this study (the taxonomic status
of its subspecies is not well-defined), in this study we did not look at substrate coloration, but rather
at the Cp of substrates. We initially considered to add "camouflage" as a standalone category
following Arbuckle and Speed (2015). However, in this study we decided to not include "camouflage"
as a category, because the analyzed image sources do not always clearly display the background on
which the animals are thriving. Therefore, we are very cautious when talking about camouflage in
the manuscript and we speculate that camouflage may be involved in shaping dorsal brightness
based on our findings. In lines 270-276 of the revised article we write: "Indeed, we found that
diurnal and lower altitude species are more likely to have a brighter integument than nocturnal and
higher elevation animals on any given substrate. Darker integuments may contribute not only to
better camouflage, but also to thermoregulatory functions in colder environments (e.g. allowing the
brain to reach optimal temperatures faster; Shine & Kearney, 2001) and UV protection (e.g.
protecting the animal from higher radiations at higher altitudes; Reguera et al., 2014)."

Specific comments.

1. Line 21. Replace belly with ventral.

Replaced

2. Line 22. Primary factor is substrate type, but an additional 2 variables are included. I would revise
the sentence. Also, why leave out the potential explanation of background matching?

We revised it as following in lines 19-20 of the revised article: "We found that substrate type,
alongside latitude and body mass, strongly influences ventral brightness". As clarified above, we did
not considered camouflage as a category, and we are not analyzing colors (i.e. hues), but rather
brightness.

3. Lines 25 – 26. Revise to say that the analysis suggest the ancestral condition was...

Revised, now it reads in lines 22-23 of the revised article: "Ancestral estimation analysis suggests
that the ancestral ventral condition was likely moderately bright".

4. Line 26. Potentially replace medium-bright with intermediate-bright.

We opted for moderately bright.

5. Line 30. The analysis does not provide details about colors only levels of brightness. Revise.

We clarified the terminology issue between color and brightness in the overview paragraph of this
document and throughout the main text, and we now talk about only brightness. We specifically
changed in lines 24-25 of the revised article to "...enhancement of ventral brightness".

6. Line 46. Replace colored tissues with pigmented integument.

**Replaced.**

7. Line 62. I would also cite Porter and Gates, since these authors described in detail the heat
balance of an ectotherm.

**We added this citation.**

8. Lines 71 – 95. The paragraph discusses melanin, which is involved in darkening rather than
coloration. This argues for a change in the title.

**We changed the title accordingly. From: “Substrate thermal properties influence ventral color
evolution in ectotherms” to: “Substrate thermal properties influence ventral brightness evolution in
ectotherms”**

9. Line 78. Should be Migliaccio.

**Corrected.**

10. Line 95. The paper focuses on brightness, but how would one classify a Viper that had a
complete ventral color of red? The authors need to provide details about how chroma is embedded
in their brightness index.

**Brightness is the relative darkness of a specific color. Two-or-more different colors (i.e. with
different hues) may have the same level of brightness. Given this definition, coloration (i.e. the hue)
is not going to affect the brightness index.**

11. Line 100. Does background matching fall under the category of camouflage?

**We initially considered to add “camouflage” as a standalone category following Arbuckle and Speed
(2015). However, in this study we decided to not include “camouflage” as a category, because the
analyzed image sources not always clearly display the background on which the animals are thriving.
Therefore, we are very cautious when talking about camouflage in the manuscript and we speculate
that camouflage may be involved in shaping dorsal brightness based on our findings. In lines 270-276
of the revised article we write: “Indeed, we found that diurnal and lower altitude species are more
likely to have a brighter integument than nocturnal and higher elevation animals on any given
substrate. Darker integuments may contribute not only to better camouflage, but also to
thermoregulatory functions in colder environments (e.g. allowing the brain to reach optimal
temperatures faster; Shine & Kearney, 2001) and UV protection (e.g. protecting the animal from
higher radiations at higher altitudes; Reguera et al., 2014).”**

12. Line 108. Appropriate citation for Reptile Database is:

Uetz, P. Freed, P. and Hošek, J. 2019 The Reptile Database <http://www.reptile-database.org>. access
[add date]

**We formatted the citation as suggested.**

13. Lines 109 – 111. Using photos/images from multiple sources is problematic. There is substantial

processing of images if one is using jpegs. Please be clear about the types of image files used, the
resolution, and file type (jpeg, raw).

We clarified and address this remark in the overview of this document and in lines 131-170 of the
revised article. In addition to that, all image formats were either jpeg, jpg, or png.

14. Line 140. Is the index used color or simply brightness?

Simply brightness. We do not analyze coloration (i.e. hue) in this study.

15. Line 143, Figure 2A. I would change the color scale to make brighter viperids as red (indicating
hotter substrates) and darker viperids green (corresponding to cooler substrates).

In addition, the scale in Figure 2A shows that *Crotalus* was not much different in color than the
ancestral condition.

We agree that the change of color scale would be beneficial, however, color blind readers may have
significant problems to visualize different shades between red and green. Hence, we did not modify
the figure.

We agree and we adapted the text adding "...few species of *Crotalus* sp." in line 212 of the revised
article and it now reads: "During the mid-late Miocene (~ 14-6 Mya), brightness of *Echis* sp. -
*Cerastes* sp. (64% [52,76], *Pseudocerastes* sp. - *Eristicophis* sp. (64% [52,76], *Bitis* sp. (61% [50,72]),
*Causus* sp. (58% [46,70]), *Daboia* sp. (58% [47,69]), and few species of *Crotalus* sp. (58% [51,64]),
independently increased (Figure 2A)..."

16. Line 141 – 147. The 95% confidence intervals overlap all the mean values presented, which does
not support an evolutionary trend. I regard this as evidence of status. Why not test for phylogenetic
signal before engaging in these analyses?

We initially tested for the phylogenetic signal, and due to a lambda = 0.75 with $p < 0.001$, we
accounted for the phylogeny in all our models.

Secondly, our aim is not to provide evidence that two genera are significantly different from each
other. On contrary, we wanted to show that, convergently, several genera enhanced and decreased
their brightness. Our ancestral estimation analyses show that different genera independently
enhanced and decreased ventral brightness over time, suggesting convergent evolution. These
findings would be meaningless if not coupled with the following MCMCglmm models, where the
results are accounting for the phylogeny and they support the evolutionary trend observed in the
ancestral estimation analysis. We have to always keep in mind that an ancestral state reconstruction
is an estimation analysis, and not a statistical test. The results from the models are backing up the
ancestral state estimations.

17. Lines 162 – 167. I am confused about the description of the pattern. Based on the formula used
in the analysis presented in the Supplementary Materials, the model was $Ventral \sim -1 + Substrate +$
$Dorsal + Altitude + Distribution + Log\ body\ mass + Polymorphic + Day\ cycle$. Substrate type does not
positively influence body mass, it should read that Ventral coloration has a negative association with
Substrate, but a positive association with body mass.

We fully agree and we corrected this phrasing; now, in the revised article lines 174-176, it reads:
"Specifically, ventral brightness, in addition to have a positive association with body mass and a
negative one with distribution variables, is strongly negatively associated with substrate type (Table
1, Table S.4-7)"

18. Line 169. I would substitute "elevation" for "altitude"

We substituted it as suggested.

19. Lines 168 – 170. Is the interpretation that vipers inhabiting light soils have brighter backs? What
about vipers that inhabit red soils and their dorsal color is red. Will the analysis verify this
correlation?

Partially correct, vipers inhabiting low cp substrates (we never talk about brightness of substrates
but rather as specific heat capacity of substrates) have relatively brighter backs. Relatively because
from Figure 4 of the revised article and from Table S.8,13 of the revised supplementary text, we can
see that ventral brightness is predicted to be much brighter within all substrate types. This is also
written in the revised article in lines 179-181: “Dorsal brightness is lower than the ventrum (Table
S.12, Figure 4b) and it is negatively associated with substrate type, but, unlike the ventrum, also with
activity pattern and altitude (Table 1, Table S.9-13, Figure 4b)”. And also in lines 266-267: “Dorsal
and head brightness were also negatively associated with the substrate type, but they were darker
than the venters across all substrate types (Figure 4)”.

As for the coloration, we hope that now we clarified that this study does not investigate coloration
(i.e. hue level), but rather brightness.

20. Lines 260 – 261. Should be “to attain body temperatures that maximize performance.”

We apologize but we cannot understand this suggestion in lines 260-261 of the original article. It
reads as: “Brightening of the dorsum and head may have been a response to the gradual
replacement of forests with open areas since 20 Mya ca (Janis et al., 2004). Brighter colors may have
been selected for camouflage and thermoregulatory purposes due to the opening of the canopy.”
Perhaps this suggestion was meant for lines 268-269? Original article reads: “Ectotherms are highly
susceptible to the surrounding environmental conditions to reach their optimal performance”. If
that is correct, we included the suggestion in lines 319-321 of the revised article and it now reads:
“Ectotherms are highly susceptible to the surrounding environmental conditions to attain body
temperatures that maximize performance.”

21. Line 290. Uetz et al. havd 365 species.

We included this in the revised article.

22. Avoid starting a sentence with “This”.

We avoided the use of “this” at the beginning of a sentence as much as possible, and now there are
only two instances throughout the revised article.

23. Line 293. Replace “path” with “history”

Replaced.

24. Line 294. Replace with “ventral colors evolved under divergent selective environments”

Replaced with the substitution of “color” with “brightness”. In the revised article it now reads (lines
352-353): “ventral brightness evolved under divergent selective environments.”

Supplementary Tables.

25. Line 300. The Google.docs link does not work.

We apologize for the accessibility issue. We tested again the link via four different laptops running
different operative systems and browsers without any problem. Could you please try the link again?
"<https://drive.google.com/drive/folders/15UMrWPMarHwDcz9GJ-i0sqWArdXAlcZ?usp=sharing>"
(perhaps a copy-paste of the last dot may have caused the problem).

26. Line 327. I would suggest replacing altitude with elevation.

Modified accordingly.

27. Line 348. How were files with different image characteristics standardized.

We hope that the explanation provided in the overview section at the beginning of this document,
and in the revised article (lines 131-170) are now addressing this concern.

28. Line 366. More details about the phylogenetic tree should be provide. How many taxa, was it
ultrametric, etc.

We added more information and in the revised article, lines 449-451, it reads: "We used the species-
level viper phylogeny by Alencar et al. (2016), which is based on 11 genes (six mitochondrial and five
nuclear) and 1186 sequences from 263 taxa, and to date is the most complete reconstruction of this
family".

29. Line 376. "contmap" is not a Bayesian ancestral reconstruction algorithm. I would use

"anc.Bayes".

Using an uninformative prior in anc.Bayes should produce concordant estimates with the ML fastAnc
function (returned by contMap). We tested this following the proposed suggestion and the below
output shows comparable estimates between the two algorithms. As we do not have prior beliefs
(i.e. no prior knowledge) when it comes to ancestral reconstruction, we believe it is better to keep
the output of the contMap reconstruction algorithm.

We also clarified in lines 453-455 of the revised article that only the multinomial analyses employed
a Bayesian approach.

30. Line 399. Explain the selection of priors.

We added to lines 483-485 of the revised article the following explanation: “As we did not have a
prior knowledge of how brightness is related to different substrates and because all models
converged (see below), we defined the default priors (Hadfield, 2018)”. As we cannot a priori
support our beliefs, and because the chain convergences were not affected by the weak priors, we
proceeded with the default “uninformative” priors.

31. Grether et al. 2004 is not cited in the text.

We removed the citation from the reference list.

32. Line 501. Italicize species names.

Italicized.

Line 567. There is a weird character in Ollala-Tarraga’s name.

Unfortunately we cannot see any issue with any of the characters of this author. Could it be the
operative system does not read this character “á”? This one: <https://en.wikipedia.org/wiki/%C3%81>
Anyhow, we now write in the the revised article line 674: “Olalla-Tarraga...”

Tables S3, S8, S 13, S18. What do the numbers in the first column represent? Add a column header
that informs the reader the meaning of the numbers.

We added in the revised supplementary text a column header to all those tables, and explained that
those numbers in the first column represent the model number from the output of the MCMCgImm
chains.

Tables S5, S6, S7, S10, S11, S12, S15, S16, S17, S20. Rather than using Substrate A, Substrate B, etc.

Please provide the actual level, i.e., arid, forest, and so on. Otherwise indicate in the table legend
what the substrate levels represent.

We indicated in all table captions of the revised supplementary text what the substrate levels stand
for.

**Reviewer #3 (Remarks to the Author): Herpetology, thermal biology**

I read the manuscript “Substrate thermal properties influence ventral color evolution in ectotherms”
by Goldenberg et al. In general, I liked the manuscript, it flows when reading and the nature of the
analyses are adequate. However, I found some aspects difficult to buy or I may have misunderstood
part of the methods. I have to confess I am not a native English speaker; therefore, I will not correct
language usage, but I marked some typos anyway.

Major Concerns.

The prediction made (line 100) looks vague or at least, it seems to pay attention to substrate only,
there is no specific mention of elevation, which was analyzed and discussed as well as other abiotic
and biotic aspects. Please, could you be more specific in the predictions in line with your results and
discussion section?

We agree and we added the predictions in lines 98-107 of the revised article: “Moreover, as latitude
in part determines how much of the sun’s radiation is received by the organism, while also directly
affecting transmission of energy to the substrate (Barry & Chorley, 2003), we expected higher
latitude species to express a greater melanic ventral integument than lower latitude counterparts.
Furthermore, body size can have a strong effect on the overall thermal inertia of an organism (Olalla-
Tárraga & Rodríguez, 2007), therefore we advanced that larger species will benefit from a brighter
venter given the slower cooling rates relative to smaller species (Moreno Azócar et al., 2016). Finally,
we predicted that high altitude species will display a darker ventral integument than low altitude
organisms, as high altitudes are generally colder than lower elevations, and a more melanic venter
may confer a thermal advantage”.

Other major concern, perhaps the more difficult to digest is the scaling of “predictor” variables. For
example, as far as I understood the types of Cp surfaces were coded. If so, the scale between
different types of Cp substrates are equally separated and this may not be the actual case. I looked
at the scale (Table S.1.) of different Cp substrate or materials and the definition of each one of the
different “categories” you used. It might be possible that the values for sandy rocky grounds could
be 0.22 and for arid substrates in combination with forest grounds and/or grass patches might be
0.30 (just guessing). Additionally, I can’t see what would be the actual difference in Cp between
arboreal (high vegetation) species and forest grounds, in any case the distance in terms of Cp
between substrate A and B does not seem to be the same than that of E and F (Lines 319-323). I
understand that getting the actual Cp values for each species is an impossible task, but I would
like to see a cautionary comment on this respect at least.

In order to make it more understandable, we reviewed the entire structure of Table S.1 in the
revised supplementary text and provided specific information, with examples, on how the categories
are defined and in turn how the scaling effect has been delineated. Also we provided more
information in lines 380-385 of the revised article: “Some categories may display similar or

overlapping substrate c_p values. For example, category B and D both present grass patches and/or
forest grounds. However, species falling within category B will likely have to endure higher ground
heat stresses than those on category D, because they also thrive on arid substrates. For a detailed
description of our substrate classification, please see Table S.1”.

To specifically address the concern about the difference in C_p between arboreal (high vegetation)
species and forest grounds, and the subsequent concern about the distance in terms of C_p between
substrates, we wrote in the revised supplementary text (lines 14-32): “We categorized substrates
based on literature information (provided in the supplied repository) on habitat type and ecology of
species. Our classification reflects biologically relevant substrates on which the studied species can
be found.”

Despite category E (high vegetation) and F (forest grounds) display similar substrate specific heat
capacity (c_p) values, the species ecology and the environmental pressures individuals experience on
those substrates are likely different. Therefore, for this analysis we separated those two categories.
Category B includes any of the arid substrates falling within category A plus any substrate falling
within category D. Category C includes at least four different substrate categories. We classified it
between category B and D because no matter which category combination is considered, the overall
heat stress from the ground is lower than the hotter substrates (i.e. A-B) and higher than the cooler
ones (D-F). A reader can argue that category C is redundant and should be incorporated within e.g.
category B or D. However, the selection pressures species have to endure in multiple substrates are
likely different from those who thrive on limited amounts, hence the addition of category C. As an
example, *Protobothrops mucrosquamatus* can be found on grasslands, forest grounds, shrublands,
around human settlements and agricultural lands, and have arboreal tendencies, thus we classified it
within category C (<https://www.iucnredlist.org/species/178409/7540882>;
<http://www.toxinology.com/fusebox.cfm?fuseaction=main.snakes.display&id=SN0111>).
We followed Dehgan, B. (2014) 5 m threshold height definition for a tree, whereas shrubs are
generally having a height of 1-5 m and multiple trunks”

The third major concern I have is with respect the authors used elevation. I see lots of overlaps and I
find difficult to separate these values to understand what was done. I looked at the Figure S.10. and I
see that some heights are forming kind of clusters, but I do not know how the authors used them in
the analyses. My guessing is that the authors used the categories in lines 332-334 that lead me to
think again about the amount of overlapping. Could the authors make this issue clearer?

We clarified the description of the altitude categorization by adding the reasoning of the selected
altitude thresholds and by providing a direct example targeting the overlapping concern. In lines
389-410 we write in the revised article: “*Altitude range*: as species are distributed across a vast
altitude range and there is very little information available on density distribution across the species
ranges, we followed the thresholds provided in the mountain system classification of Körner et al.
(2005), and produced the following five categories: Low ($x \leq 500$ m), Low-Medium ($x \leq 1000$ m),
Medium-High ($500 \text{ m} < x \leq 4000$ m), High ($1000 < x \leq 4000$ m), All (all the range). In this study, the
threshold of 4000 m corresponds to the highest examined viper distribution (i.e. *Gloydius halys* –
Tuniyev et al., 2009). Furthermore, vegetation-based zonation at lowlands are very susceptible to
climate conditions (Salter et al., 2005), thus different regions experience different upper limits for
vegetation zonation, in turn potentially affecting the heat exchange dynamic between substrate-
organism. Therefore, we increased the lowland elevation from 300 m (proposed by Körner et al.
(2005)) to 500 m as the latter reflects an average of the lowland thresholds (upper elevation limits
for lowland zonation are 600-700 m - Couplan & Ligeon, 2005). Finally, some viper species can span
across a vast altitudinal range (e.g. *Atropoides picadoi* - Solórzano et al., 2014), while others are

restricted to a well-defined zone (e.g. *Bothriechis rowleyi* - Canseco-Márquez et al., 2007). Therefore,
a vastly distributed species may overlap the distribution of a limitedly distributed one. However, the
species that spans across a vast range will likely be exposed to more selection pressures due to
different levels of abiotic factors, such as humidity, temperature and solar radiation, that can
ultimately affect the species thermal balance. Consequently, in this study we produced the above-
mentioned altitude categories that also account for the species' ecology. A graphical representation
of altitude distribution across species is in Figure S.10".

Lastly, and understanding the complexity of the analyses carried out, I feel that despite the
quantitative value of these variables (Cp and elevation) they are not independent of climatic
variables, such as temperature. In my opinion even when sand Cp may be the same in light of the
physical characteristics of sand; it is not the same a sandy cold desert (Patagonia, for example) than
the Sahara sands in terms of heat dissipation for a snake. Could the authors include a paragraph
considering this type of information?

We fully agree with this concern, and we clarified in lines 410-419 of the revised article how
including latitude as variable can "control" the climate for both Cp and elevation: "*Distribution: the*
*studied species are distributed across all latitudes, but, similarly to elevation range, very little*
*information is available on density distribution across the species ranges. Therefore, we classified*
*the latitudinal distribution following the radiation index presented in Barry & Chorley (2003):*
*Tropical, Subtropical, Temperate, Temperate-Polar or any combination thereof. The level*
*Temperate-Polar presented only one species (*Vipera berus*), however we kept that in our analyses as*
*it did not affect model convergence. Finally, latitude is linked with solar radiation, in turn affecting*
*temperature. Our models, by including the latitudinal distribution of a species, account for the effect*
*of different solar radiation levels, and thus, indirectly, temperatures, on shaping the physical*
*characteristics of the local substrate".*

With the example of Patagonia and Sahara, despite the two substrate Cps are falling within the same
category, the latitudes are different, therefore the solar radiation hitting the local substrate is going
to be different, and the model will account for this difference and will not see the two locations
having the same substrate Cp effect on the integument brightness.

As for the altitude, let's assume a Low-Medium elevation ($x \leq 1000$ m) on the Kilimanjaro and a Low-
Medium elevation on the European Alps. Both zones will fall within the same Altitude category,
however the solar radiation at the Kilimanjaro will be higher than at the European Alps, due to the
difference in latitudes, in turn potentially affecting the surrounding temperature and substrate
thermal properties. Thus, similarly to the previous example, by including latitude, the model will
account for this variation.

I really liked the way the authors used color information from different sources and how they
managed to input these data in a trustable and objective way. I also enjoyed the arguments about
the electrical properties of melanin. I can see the authors really manage the subject.

Minor aspects.

Lines 118-127. For is written in boldface, should be regular font.

Corrected.

Line 136 Vis instead of vis.

Corrected.

I am not totally sure about the need of Figure 1. I might be wrong. Explain why the Figure must stay.

We agree and we moved Figure 1 of the original article to the revised supplementary text. Instead

we provide now in Figure 1 of the revised article a diagram showing the schematic of our main

research question (i.e. the link between thermal radiation and dark-bright ventral integument).

Revise references, there are some species names not in italics.

We went through the references and we addressed this comment.

As mentioned before, I liked the way the manuscript is written and the focus the authors gave to it. I

hope the authors could fix the major issues or at least make reasonable arguments to keep them

and to explain the readers the limitations of using categories in the case of Cp substrates and

elevation.

Reviewers' comments:

Reviewer #1 (Remarks to the Author):

I think that the authors have done very well in responding to my initial comments, and I don't have any further criticisms of the revised manuscript. Nice paper, well done.

Reviewer #2 (Remarks to the Author):

Review of "Substrate thermal properties influence ventral brightness evolution in ectotherms"

How does the thermal properties of the substrate influence the ventral brightness of ectotherms? This question is the focus of the current study. The manuscript has been revised to accommodate the suggestions made by three reviewers. I commend the authors for their thoughtful responses to the reviewers comments and the modifications and additions that improved the paper. Although the revised manuscript is much improved, there remain a few issues that should be addressed.

Line 22. I am unclear as to how circadian rhythms can be a selective force. Activity pattern, i.e., diurnal, nocturnal, or cathemeral, could influence how strong an affect substrate temperature would influence body temperature. As a consequence, diurnal species are predicted to be more sensitive to variation in Cp. I would revise this statement.

Lines 25 – 26. I could not find the evidence supporting the statement that ventral brightness affects the behavioral ecology of ectotherms.

Line 46. Change to "..., the pigmented tissue can directly affect body temperature" Delete "the"

Line 52. Porter, W. P., Mitchell, J. W., Beckman, W. A., DeWitt, C. B. 1973. Behavioral implications of mechanistic ecology - Thermal and behavioral modeling of desert ectotherms and their microenvironment. *Oecologia* 13: 1-54 should be cited in figure 1.

Line 68. Delete "This is particularly true because" and start the topic sentence of the paragraph with "Brightness of the integument..."

Line 79. I suggest revising to "hot radiative and conductive substrates..."

Line 81. "focused on dorsal coloration..."

Line 84. "ecological significance of ventral coloration..."

Line 105. I would suggest changing "altitude" with "elevation". The two are not interchangeable.

Line 129. I suggest moving the section "Study design and methodology validation" to the Methods section.

Line 130. What is a "well-described" species. Consider deleting the phrase.

Line 151. I think it is sufficient to express the p-value as <0.0001 . I find the use of exponential notation distracting.

Line 161. Energy-rich.

Lines 173 – 174. I do not think it is correct to say the environmental and morpho-behavioral variables are uncoupled, because there is no test of the interaction among those predictor variables. It is reasonable to suggest that brightness across different regions of the body are

influenced by non-overlapping predictor variables.

Line 175. Why not use latitude in place of distribution?

Figure 4 should be Figure 2.

Line 180. Is activity pattern the behavioral trait of the study?

Line 182. "more likely to be brighter..."

Line 185 and Figure S10b. It is not clear what trait was used to assess the correlation between body regions. I would use lower case "r" for the correlation coefficient.

Line 207. Figure 2 should be figure 3.

Line 209. The statement that several groups exhibit convergence should be tested using the R package "convevol." The pattern in Figure 2A does not appear to provide strong support for convergence in brightness.

Line 237. "...species living on arid substrates..."

Line 238. I would add a sentence or two describing the mechanism of heat dissipation. Could it be possible that a brighter venter also reduces conduction of heat?

Line 268. Replace circadian rhythm with activity pattern or activity period.

Table 1. I would replace "Day Cycle" with "Activity Pattern" or "Activity Period". Polymorphism is not defined in the table legend.

Lines 375 – 380. In reviewing the classification scheme for Cp, was there an attempt to demonstrate each category was significantly different from one another. One approach is to use "gap coding". Based on the values presented in Table S1, it looks as though there is overlap in Cp between some categories. Alternatively, one could construct a histogram to determine if the Cp values in each category are discrete and show limited overlap.

Line 384. I would replace "thrive" with exploit or something similar.

Line 404. I would suggest using "broadly distributed" rather than "vastly distributed."

Reviewer #3 (Remarks to the Author):

I reviewed the manuscript "Substrate thermal properties influence ventral brightness evolution in ectotherms" by Goldenberg et al. In my previous review I found the manuscript interesting and I liked the way the authors treated the topic. I found it original.

In this new version, I consider the authors fulfilled my expectations. As mentioned before, the writing flows correctly, despite the complexity of the subject that needs a great amount of supplementary material.

The authors answered all my concerns in a proper way and their explanations were reasonable. After this, I find this new version acceptable for publication.

**Overview**

List of terms used in this document:

**Original article:** the main article submitted during revision #1

**Revised article:** the original article revised

**Original supplementary text:** the supplementary file submitted during revision #1

**Revised supplementary text:** the original supplementary file revised

We thank Reviewer #2 for the insightful comments and Reviewers #1 and #3 for their approvals. We
thoroughly analyzed all the concerns and addressed them point-by-point. All changes in the revised
texts (main article + supplementary) are highlighted in yellow, as well as our answers in this
document.

As a further support to our findings, in the latest revised article we cite in lines 271-273 a recent
study (Martínez-Freiría et al. 2020) that provides another line of evidence for the advantage of dark
integument colors in cold environments.

**Reviewer #1** (Remarks to the Author):

I think that the authors have done very well in responding to my initial comments, and I don't have
any further criticisms of the revised manuscript. Nice paper, well done.

**Reviewer #2** (Remarks to the Author):

Review of "Substrate thermal properties influence ventral brightness evolution in ectotherms"

How does the thermal properties of the substrate influence the ventral brightness of ectotherms?
This question is the focus of the current study. The manuscript has been revised to accommodate
the suggestions made by three reviewers. I commend the authors for their thoughtful responses to
the reviewers comments and the modifications and additions that improved the paper. Although the
revised manuscript is much improved, there remain a few issues that should be addressed.

1) Line 22. I am unclear as to how circadian rhythms can be a selective force. Activity pattern, i.e.,
diurnal, nocturnal, or cathemeral, could influence how strong an affect substrate temperature would
influence body temperature. As a consequence, diurnal species are predicted to be more sensitive to
variation in Cp. I would revise this statement.

We agree with the reviewer that our previous wording required clarification. Our analyses on the
dorsal brightness show that diurnal and lower altitude species are more likely to be brighter than
nocturnal and higher altitude animals. Therefore, the daily light periodicity can be a selective force.
we replaced the words "driven by" with "associated with", and we added the variable "altitude". The
sentence in the revised article now reads as: "Substrate type also significantly affects dorsal
brightness, but this is associated with different selective forces: activity-pattern and altitude."

2) Lines 25 – 26. I could not find the evidence supporting the statement that ventral brightness
affects the behavioral ecology of ectotherms.

As correctly pointed out in comment #17 below, activity pattern is a behavioral trait. Thus, our
findings suggest an association between morphological and behavioral traits and ecological
pressures all within an evolutionary framework. Nevertheless, we reworded the sentence in abstract
and now reads in lines 25-27 of the revised article: “We provide evidence that integument
brightness can impact the behavioral ecology of ectotherms”.

3) Line 46. Change to “..., the pigmented tissue can directly affect body temperature” Delete “the”

Replaced as suggested.

4) Line 52. Porter, W. P., Mitchell, J. W., Beckman, W. A., DeWitt, C. B. 1973. Behavioral implications
of mechanistic ecology - Thermal and behavioral modeling of desert ectotherms and their
microenvironment. *Oecologia* 13: 1–54 should be cited in figure 1.

We adapted the diagram from Porter and Gates (1969) and Porter et al. (1973). As suggested, we
now added in the caption of Figure 1 the citations, as well as in the references list.

5) Line 68. Delete “This is particularly true because” and start the topic sentence of the paragraph
with “Brightness of the integument...”

We removed “This is particularly true because”.

6) Line 79. I suggest revising to “hot radiative and conductive substrates...”

We revised as suggested.

7) Line 81. “focused on dorsal coloration...”

Replaced.

8) Line 84. “ecological significance of ventral coloration...”

Replaced.

9) Line 105. I would suggest changing “altitude” with “elevation”. The two are not interchangeable.

Replaced as suggested.

10) Line 129. I suggest moving the section “Study design and methodology validation” to the
Methods section.

We initially considered placing this section in Material and Methods. However, we believe that the
validation of such methodology is a result *per se*. Therefore, we decided to keep this section in
Results.

11) Line 130. What is a “well-described” species. Consider deleting the phrase.

We meant taxonomically well-defined species. In the revised article at line 116 it now reads as: “We
analyzed 126 taxonomically unambiguous viper species from 31 genera...”

12) Line 151. I think it is sufficient to express the p-value as <0.0001 . I find the use of exponential
notation distracting.

In the revised article, line 151 (in the current revised article line 136), we now express the p-value as
suggested. As well as the following p-values in lines 137, 141, 150, and 154.

13) Line 161. Energy-rich.

Revised as suggested.

14) Lines 173 – 174. I do not think it is correct to say the environmental and morpho-behavioral
variables are uncoupled, because there is no test of the interaction among those predictor variables.
It is reasonable to suggest that brightness across different regions of the body are influenced by non-
overlapping predictor variables.

We agree and in the revised article, lines 159-160, it now reads as: “However, environmental and
morpho-behavioral variables differ depending on body region.”.

15) Line 175. Why not use latitude in place of distribution?

Replaced as suggested.

16) Figure 4 should be Figure 2.

Thank you for pointing this out. We modified the numbers of figure sequences accordingly.

17) Line 180. Is activity pattern the behavioral trait of the study?

Correct. We clarify this in comment #2, and we rephrase our answer:

We write this in lines 387-390 of the revised article: “We also incorporated other environmental
data that could significantly affect the amount of the sun’s energy-rich radiation received by the
animal and substrate^{65,66}, and morpho-behavioral information that can influence thermal heat
transfer (for a frequency table please see Table S.22): ...”; and it continues in line 420-422: “...5)
*Activity pattern*: behavioral trait defined as Diurnal, Nocturnal, Both or Unknown (all references used
to score and classify the analyzed variables are available at the provided repository).”. We added
“behavioral trait defined as” to strengthen this point.

18) Line 182. “more likely to be brighter...”

We corrected that as suggested.

19) Line 185 and Figure S10b. It is not clear what trait was used to assess the correlation between
body regions. I would use lower case “r” for the correlation coefficient.

We assessed the correlation using brightness levels of dorsum and head regions. We now clarified
this in the revised article and it reads in lines 170-171 as: “The result is not surprising, as the
brightness levels of two regions are strongly associated with each other (Figure S.10B)”.

20) Line 207. Figure 2 should be figure 3.

We addressed this issue following comment #16.

21) Line 209. The statement that several groups exhibit convergence should be tested using the R
package “conevol.” The pattern in Figure 2A does not appear to provide strong support for
convergence in brightness.

We thoroughly looked into the R-package “conevol”, specifically into the functions “convnum”,
“convnumsig”, “convrat”, and “convratsig” and we read the publication associated with this package
(Stayton 2015). We now provide the Stayton’s convergence metrics C1-C5 (Table S.23) and a
polymorphospace showing convergent evolution of the focal taxa (see below and Figure S.14). We
describe these results in lines 201-207 of the revised article supported by Table S.23 and Figure S.14
in the revised supplementary text. We also added these analyses to the methodology section in lines
459-472 of the revised article and the associated citations to the references list. In the results
section, lines 201-207 of the revised article, it reads as: “The results are further supported by the
Stayton’s C1–C5 metrics of convergences based on dorsal and ventral brightness on the focal taxa
(Table S.23); specifically C1 (i.e. the maximum distance between two lineages that has been brought
together by subsequent evolution) = 0.54 ($p = 0.00$) coupled with C5 (i.e. the number of convergent
focal taxa that reside in a distinct region of the polymorphospace) = 15 ($p = 0.00$) show that the focal
taxa significantly cluster together in a separate region of the polymorphospace driven by ventral
brightness (Table S.23, Figure S.14).”.

Moreover, to further support our findings, we now provide a phenogram with 95% C.I. in the revised
supplementary text (Figure S.15B and here below) showing that ventral brightness shifted to bright
brightness levels independently over time across different genera – result reported in the revised
article in lines 207-210 : “As another line of evidence, the 95% C.I. phenogram, which projects the
phylogeny in a space defined by the ventral brightness over time (Figure S.15B), shows that ventral
brightness of the focal taxa shifted to bright brightness levels independently over time,
corroborating our findings.”.

**Figure S.14.** Polymorphospace output from “convevol” defined by dorsal and ventral brightness on the 126
 studied viper species showing convergent evolution for the focal species (for ventral brightness): *Bitis*
 *parviocula*, *Bitis peringueyi*, *Causus resimus*, *Causus defilippii*, *Daboia mauritanica*, *Eristicophis macmahoni*,
 *Pseudocerastes urarachnoides*, *Pseudocerastes persicus*, *Cerastes cerastes*, *Cerastes gasperettii*, *Echis*
 *pyramidum*, *Echis omanensis*, *Echis coloratus*, *Echis leucogaster*, *Crotalus ruber*, *Crotalus cerastes*. Full black
 dots denote non-focal species or nodes, and partially full dots denote focal species. Red arrows indicate the
 nodes/lineages that cross into the region of convergent species. The purple area defines the region where the
 focal taxa are present. The Stayton’s metrics supporting this output are reported in Table S.23.

**Figure S.15. a)** Ancestral state estimation of the ventral brightness contrasted with a **b)** Phenogram with 95%

confidence intervals projecting the phylogeny in a space defined by the ventral brightness and time showing
that bright ventral colors evolved independently over the vipers' evolutionary history; diverse colored-bars
represent different genera. Only the first 16 brightest species have been colored for graphical interpretation.

22) Line 237. "...species living on arid substrates..."

We replaced "thriving" with "living" as suggested.

23) Line 238. I would add a sentence or two describing the mechanism of heat dissipation. Could it
be possible that a brighter venter also reduces conduction of heat?

We describe in lines 248-250 the mechanisms of heat dissipation, and we believe it would be a
repetition to add the suggested point to line 238 (now 239). Moreover, the thermal properties of
melanin are well studied and, to a lesser degree, its conductive properties. On the contrary, the
thermal and conductance properties of iridophores and their overall effect on the thermal balance
of an organism are virtually unknown. Therefore, we did not speculate much about this.

24) Line 268. Replace circadian rhythm with activity pattern or activity period.

We replaced it with "activity pattern".

25) Table 1. I would replace "Day Cycle" with "Activity Pattern" or "Activity Period". Polymorphism is
not defined in the table legend.

We replaced "Day Cycle" with "Activity Pattern" and we added the definition of Polymorphism in the
table legend as suggested.

26) Lines 375 – 380. In reviewing the classification scheme for Cp, was there an attempt to
demonstrate each category was significantly different from one another. One approach is to use
"gap coding". Based on the values presented in Table S1, it looks as though there is overlap in Cp
between some categories. Alternatively, one could construct a histogram to determine if the Cp
values in each category are discrete and show limited overlap.

Yes, we took into consideration this point and we are aware of it. In the previous revision round,
Reviewer #3 raised this concern of categories overlapping, and we addressed that as following:

In lines 382-386 of the revised article: "Some categories may display similar or overlapping substrate
c_p values. For example, category B and D both present grass patches and/or forest grounds.
However, species falling within category B will likely have to endure higher ground heat stresses
than those on category D, because they also thrive on arid substrates. For a detailed description of
our substrate classification, please see Table S.1".

To specifically address the concern about the difference in Cp between arboreal (high vegetation)
species and forest grounds, and the subsequent concern about the distance in terms of Cp between
substrates, we wrote in the revised supplementary text (lines 14-32): "We categorized substrates
based on literature information (provided in the supplied repository) on habitat type and ecology of
species. Our classification reflects biologically relevant substrates on which the studied species can
be found.

Despite category E (high vegetation) and F (forest grounds) display similar substrate specific heat
capacity (c_p) values, the species ecology and the environmental pressures individuals experience on

those substrates are likely different. Therefore, for this analysis we separated those two categories.
 Category B includes any of the arid substrates falling within category A plus any substrate falling
 within category D. Category C includes at least four different substrate categories. We classified it
 between category B and D because no matter which category combination is considered, the overall
 heat stress from the ground is lower than the hotter substrates (i.e. A-B) and higher than the cooler
 ones (D-F). A reader can argue that category C is redundant and should be incorporated within e.g.
 category B or D. However, the selection pressures species have to endure in multiple substrates are
 likely different from those who thrive on limited amounts, hence the addition of category C. As an
 example, *Probothrops mucrosquamatus* can be found on grasslands, forest grounds, shrublands,
 around human settlements and agricultural lands, and have arboreal tendencies, thus we classified it
 within category C (<https://www.iucnredlist.org/species/178409/7540882>;
 <http://www.toxinology.com/fusebox.cfm?fuseaction=main.snakes.display&id=SN0111>).
 We followed Dehgan, B. (2014) 5 m threshold height definition for a tree, whereas shrubs are
 generally having a height of 1-5 m and multiple trunks”.

Furthermore, as suggested, below we provide a classification of the representative cp values (dots),
 reported in Table S.1 of the revised supplementary text, across the substrate categories. Lines
 express the range across which substrate categories are defined. As explained in the paragraph
 above to Reviewer #3 and in the revised article and supplementary text, it shows limited overlaps
 which reflect the ecological and environmental pressures species have to endure occupying multiple
 (or limited) substrate types. Note that category C was removed from the plot because, as previously
 explained, it includes at least four different substrate categories and no matter which category
 combination is considered, the overall heat stress from the ground is likely lower than the hotter
 substrates (i.e. A-B) and higher than the cooler ones (D-F).

 27) Line 384. I would replace “thrive” with exploit or something similar.

We replaced “thrive” with “exploit” as suggested.

28) Line 404. I would suggest using “broadly distributed” rather than “vastly distributed.”

We modified it as suggested.

Reviewer #3 (Remarks to the Author):

I reviewed the manuscript "Substrate thermal properties influence ventral brightness evolution in
ectotherms" by Goldenberg et al. In my previous review I found the manuscript interesting and I
liked the way the authors treated the topic. I found it original.

In this new version, I consider the authors fulfilled my expectations. As mentioned before, the
writing flows correctly, despite the complexity of the subject that needs a great amount of
supplementary material.

The authors answered all my concerns in a proper way and their explanations were reasonable.

After this, I find this new version acceptable for publication.